# A domain wall in twisted M-theory

**Jihwan Oh[1*] and Yehao Zhou[2]**

**1** Mathematical Institute, University of Oxford,
Woodstock Road, Oxford, OX2 6GG, United Kingdom
**2** Perimeter Institute for Theoretical Physics,
31 Caroline St. N., Waterloo, ON N2L 2Y5, Canada

⋆ jihwan.oh@maths.ox.ac.uk

## Abstract

We study a four-dimensional domain wall in twisted M-theory. The domain wall is engineered by intersecting D6 branes in the type IIA frame. We identify the classical algebra of operators on the domain wall in terms of a higher vertex operator algebra, which describes the holomorphic subsector of a 4d $\mathcal{N} = 1$ supersymmetric field theory, and compute the associated mode algebra. We conjecture that the quantum deformation of the classical algebra is isomorphic to the bulk algebra of operators from which we establish twisted holography of the domain wall.

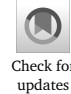
# 1   Introduction

Topological twists [1, 2] have been a standard tool to study a supersymmetric quantum field theory along with $\Omega-$deformation [3, 4]. Since the twists make the translation generator $Q$-exact with respect to the scalar supercharge $Q$ of the twisted theory, the Q-cohomology class of a Q-closed local operator is independent of its position. One can slightly generalize the notion of the topological twist and discuss the holomorphic version. In this case, the Q-cohomology class only retains the holomorphic dependence. By construction, the resulting holomorphic field theory needs to be even dimensional. Outstanding examples include the 2d $(0, 2)$ theory [5–7] and 4d $\mathcal{N} = 1$ theory [8–10]. The protected subsector of the 2d example naturally forms a familiar vertex operator algebra. On the other hand, the holomorphic sector of 4d $\mathcal{N} = 1$ theory reorganizes itself into a higher vertex operator algebra (higher VOA) [11–14], which will be the main player of this paper. Higher VOA naturally appears in twisted M-theory [15, 16], where there are 4 holomorphic directions and 7 topological directions.

Twisted M-theory [16,17] is an eleven-dimensional supergravity background specified by a product geometry of a hyperKähler 4-manifold and a 7-manifold with $G_2$-holonomy, equipped with the above mentioned twist and $\Omega-$deformation. The hyperKähler 4-manifold is equipped with the holomorphic twist and the $G_2$-manifold is equipped with the topological twist, along with the $\Omega-$deformation. Due to the $\Omega$-deformation, the M-theory degree of freedom localizes to the 5d Chern-Simons theory, which is topological in one direction and holomorphic in four directions; it comes from Taub-NUT geometry inside the $G_2-$manifold, or a KK-monopole. Once we introduce M2 and M5 branes, the worldvolume theories of M2 and M5 branes are twisted and deformed(and localized to the subspace of the spacetime). To preserve supersymmetry and to be compatible with the twisted background, the M2 branes should wrap a three-cycle in the $G_2$ manifold and the M5 branes should wrap a four-cycle in the $G_2$ manifold. The supersymmetric coupling between the membranes and the supergravity localizes to the coupling between defects and the 5d Chern-Simons theory; the M2 branes and M5 branes give rise to line and surface defects.

Twisted holography in the context of M-theory is an exact isomorphism between the algebra of operators of the 5d Chern-Simons theory and the algebra of operators on membranes. The relevant algebras are affine $gl(1)$ Yangian [18, 19] and its shifted and truncated version [20–24]. The isomorphism is induced by Koszul duality,[1] which is a common name for a duality between a certain pair of algebras. Conjecturally, twisted holography describes the protected subsectors of well-established M2 and M5 brane holography [27,28]. We will discuss the relevant background for twisted M-theory in §2.1, §2.2.

One can also consider a network of M2 and M5 branes [24,29] and identify the relevant algebra [30]. Following the idea of [17], where the author beautifully derived the M2 brane algebra from the perturbative computation in the 5d Chern-Simons theory, one can reproduce the algebra associated to the network of M2 and M5 branes [31].

---

[1]For another nice set of physical examples of Koszul duality not in the context of twisted holography, see [25,26].

Twisting supergravity is not necessarily restricted to the eleven-dimensional example. Indeed, the most complete example for twisted holography is [32] the twisted type IIB theory and a protected subsector of D3 brane holography. See also [33] as an example of a D2-D4 system and [34] as an example of a D1-D5 system. Literatures that employ direct localization in both supergravity and field theory include [35, 36].

In this paper, we study the seemingly last available BPS defect in the twisted M-theory that has not been not studied: an extra KK-monopole, which is different from the KK monopole that descends to the 5d Chern-Simons theory. The new KK-monopole creates a domain wall-like defect wrapping the 4 holomorphic directions in the 5d Chern-Simons theory. Different from the line and surface defects engineered by membranes, the domain wall-like defect is engineered by changing the $G_2$-holonomy 7-manifold [37, 38]. It passes tight constraints for the allowed BPS objects in twisted M-theory. (1) It preserves some amount of supersymmetry. (2) It is compatible with the topological holomorphic background. (3) It preserves the $G_2$-holonomy structure. (4) It preserves the $U(1)$ isometry of the $G_2$ manifold.

For a practical computation and convenient visualization, it is better to work in the type IIA frame by reducing along a circle in the $G_2$ manifold. In type IIA, we obtain two sets of intersecting D6 branes [39], one set from the original Taub-NUT$_K$ geometry and the other set from the change we made. The original $K$ D6 brane worldvolume theory gives the 5d $G = U(K)$ Chern-Simons theory and the new $N$ D6' brane intersects with the 5d Chern-Simons theory as a domain wall. On the wall, D6-D6' strings are localized and realize the 4d $\mathcal{N} = 1$ chiral multiplet [39]. Due to the holomorphic twist on the wall, we get a higher VOA. We will give more details about the intersecting D6 brane configuration in §2.3.

The major difference between the higher VOA coming from the holomorphic twist and the familiar 2d VOA is that in the former case the descent operators play the key role to generate meromorphicity, which is the essential ingredient for the algebraic structure under the vertex operator algebra. Like the usual cohomological field theory, the set of Q-cohomology classes in the higher VOA includes a certain integrated version of the descent operators and in our case they are holomorphic descents of local Q-cohomology classes. The holomorphic descents themselves are not Q-closed, but combining with the holomorphic top form of the 4d spacetime manifold $\mathbb{C}^2$ and integrating over $S^3 \subset \mathbb{C}^2$, it becomes a Q-cohomology class. We will explain in more detail about this set of operators and how it generates the higher VOA in §3.1, §3.2. Since the domain wall forms a boundary of the 5d Chern-Simons theory, we need to account for the boundary condition of the 5d Chern-Simons gauge field, which we did in §3.3. Collecting these ingredients, we propose the physical observables living at the domain wall in §3.4. The 5d/4d system that we have discussed is anomalous due to the chiral fermion at the domain wall. It can be canceled by tuning a parameter of the 5d Chern-Simons theory on one side of the wall, as we will discuss in §3.5.

In §4, we will compute the commutation relation of the physical observables that we have constructed in the previous section. We prove that the resulting algebra is the universal enveloping algebra of $\text{Diff}_{\epsilon_2}\mathbb{C} \otimes \mathfrak{gl}_K$. We conjecture that the quantum deformation of our classical computation is isomorphic to the Koszul dual algebra of operators in the 5d Chern-Simons theory. A priori, this conjecture must be true due to the powerful theorem by Costello [17]: there is a unique deformation of $U(\text{Diff}_{\epsilon_2}\mathbb{C} \otimes \mathfrak{gl}_K)$ and it is the algebra of operators in the 5d Chern-Simons theory. In this sense, the proof for the conjecture is already given. However, we prefer to call it the conjecture since we do not know how to systematically incorporate the quantum corrections only using our machinery but not relying on the Costello's theorem.

We conclude in §5 by summarizing the discussion and provide open questions that we want to answer in the future. In Appendix, we provide details of some computations that we have abbreviated in the main text. We collect relevant 4d $\mathcal{N} = 1$ supersymmetry transformation of a chiral multiplet in Appendix A. Lastly, we provide details of the key commutation relation

computation in Appendix B.

## 2 Twisted M-theory

In §2.1, we will review how to define a twisted supergravity and its deformation under the $\Omega$−background. We will focus on the 11-dimensional example, which we call the twisted M-theory. Due to the Omega background, the bulk dynamics drastically simplifies to a non-commutative version of the 5d Chern-Simons theory, which is topological in 1 direction and holomorphic in 4 directions. We will briefly explain the algebra of operators in the 5d Chern-Simons theory. In §2.2, we will explore various BPS objects in the twisted M-theory and study their compatibility with the twisted background. Finally, in §2.3, we identify the BPS object in M-theory that realizes the domain wall in the 5d Chern-Simons theory.

### 2.1 Background

Twisted supergravity [15] is defined as the supergravity in a background where the bosonic ghost $\Psi$ of the local supersymmetry takes a nonzero value. To satisfy the supergravity equation of motion, $\Psi$ needs to square to zero, i.e., it is nilpotent. In the presence of $U(1)$ isometry in the background geometry, we may turn on the $\Omega$-background for the twisted supergravity background $X$ and deform it to $X_\epsilon$. $X_\epsilon$ is equipped with $\Psi_\epsilon$, which squares not to zero but to $\epsilon V$, where $V$ is the vector field generating the $S^1$ action.

In [16] Costello defines the twisted and $\Omega$−deformed 11-dimensional supergravity and proves that it satisfies the equation of motion of the 11-dimensional supergravity. We will call this background twisted M-theory. The twisted M-theory background[2] is specified by the triple $(\Psi_\epsilon, g, C)$, which are a bosonic ghost, a metric, and an M-theory 3-form. The 11-dimensional manifold is a product of a 7-manifold $\mathcal{M}_7^T$ with $G_2$ holonomy and a hyperKähler 4-manifold $\mathcal{M}_4^H$ parametrized by $z$ and $w$. The background induced by $\Psi_\epsilon$ makes the dependence on $\mathcal{M}_7^T$ topological and the dependence on $\mathcal{M}_4^H$ holomorphic.

The background geometry is given by

$$(\mathbb{C}_z \times \mathbb{C}_w)^H \times (\mathbb{R}_t \times \mathbb{C} \times TN_K)^T. \tag{1}$$

$TN_K$ is the Taub-NUT$_K$ manifold, which can be thought of as a circle($S^1_{TN}$) fibration over the base $\mathbb{R}^3$. There is an $S^1_{\epsilon_1} \times S^1_{\epsilon_2}$ action on $\mathbb{C} \times TN_K$. The first $S^1_{\epsilon_1}$ action, parametrized by $\epsilon_1$, acts on $\mathbb{C}$ and the base of the Taub-NUT manifold simultaneously. We will therefore put the subscript $\epsilon_1$ on the $\mathbb{C}$ and denote it as $\mathbb{C}_{\epsilon_1}$. Also, we will denote the Taub-NUT base $\mathbb{R}^3$ as $\mathbb{R}^3_{TNB}$. The second $S^1_{\epsilon_2}$ action, parametrized by $\epsilon_2$, acts on the Taub-NUT circle. Finally, the $S^1 \times S^1$ action preserves the holomorphic volume form of $\mathbb{C}_{\epsilon_1} \times TN_K$.

For a practical computation, it is helpful to reduce along $S^1_{TN}$ to go to type IIA theory. Note that we do not lose essential information by the circle reduction, since it is a Q-exact operation, as proven in [16]. Under the reduction, the M-theory geometry produces $K$ D6 branes. Furthermore, due to the localization effect of the Omega background on $\mathbb{C}_{\epsilon_1}$, the 7d maximal SYM of D6 brane worldvolume reduces to the 5d Chern-Simons theory on $\mathbb{R}_t \times \mathbb{C}_z \times \mathbb{C}_w$:

$$\frac{1}{\epsilon_1} \int_{\mathbb{R}_t \times \mathbb{C}_z \times \mathbb{C}_w} dz \wedge dw \wedge \left( AdA + \frac{2}{3} A \wedge_{\epsilon_2} A \wedge_{\epsilon_2} A \right), \tag{2}$$

---

[2]Appendix A of [16] contains the exact expression of the background and the proof for it to be a consistent supergravity background.

where $\wedge_{\epsilon_2}$ is a combination of a wedge product and a Moyal product $\star_{\epsilon_2}$. The Moyal product between two holomorphic functions is defined as

$$f \star_{\epsilon_2} g = f g + \epsilon_2 \frac{1}{2} \epsilon_{ij} \frac{\partial}{\partial_{z_i}} f \frac{\partial}{\partial_{z_j}} g + \dots \tag{3}$$

Due to the holomorphic top form in the action, the 5d gauge field has only three components:

$$A = A_t d t + A_{\bar{z}} d\bar{z} + A_{\bar{w}} d\bar{w}. \tag{4}$$

The origin of the non-commutativity that induces the Moyal product is the B-field that descends from the M-theory 3-form C-field

$$C = dz \wedge dw \wedge V^\flat, \tag{5}$$

where $V$ is a vector field that generates the rotation on $S^1_{TN}$ and $V^\flat$ is the corresponding dual 1-form, whose component is given by $V^\flat_\mu = g^{TN}_{\mu\nu} V^\nu$, where $g^{TN}_{\mu\nu}$ is the Taub-NUT metric. We reduce along $S^1_{TN}$ when we pass to the type IIA frame.

The reason that we may entirely focus on the open string part(the D6 brane worldvolume theory) ignoring the closed strings from the geometry is that the B-field combines with the B-model background(holomorphic) on $\mathbb{C}_z \times \mathbb{C}_w$ and effectively produces the A-model background(topological). As we already have A-model background in the other 6 directions, $\mathbb{R}_t \times \mathbb{C}_{\epsilon_1} \times \mathbb{R}^3_{TNB}$, all 10 directions in type IIA are topological. In other words, physical closed string observables in the relevant Q-cohomology have a trivial dependence on spacetime coordinates. Hence, they can be ignored [16].

Let us discuss the algebra of operators in the 5d Chern-Simons theory: $\text{Obs}^{5d}_{\epsilon_1,\epsilon_2}$. We will first describe the case with $\epsilon_1 = 0$, $\text{Obs}^{5d}_{0,\epsilon_2}$ and turn on the deformation parameter $\epsilon_1$ later. Since the equation of motion of the 5d Chern-Simons theory is $F = 0$, all operators have a positive ghost number. The ghosts have a trivial dependence on $t$ and holomorphic dependence on $z$ and $w$. Due to the non-commutative background on $\mathbb{C}_z \times \mathbb{C}_w$, $[z,w] = \epsilon_2$, the operators form a non-commutative algebra of holomorphic functions on $\mathbb{C}_z \times \mathbb{C}_w$. Together with the BRST differential $\delta$, the algebra of operators of the 5d Chern-Simons theory forms a graded associative algebra isomorphic to

$$\wedge^* \mathbb{C}[z,w]_{\epsilon_2} \otimes \mathfrak{gl}_K \cong \wedge^* \text{Diff}_{\epsilon_2} \mathbb{C} \otimes \mathfrak{gl}_K, \tag{6}$$

where $\text{Diff}_{\epsilon_2} \mathbb{C}$ is the algebra of differential operators on $\mathbb{C}_z$ with $[z, \partial_z] = \epsilon_2$. In other words, the classical algebra $\text{Obs}^{5d}_{0,\epsilon_2}$ is the Chevalley-Eilenberg algebra $C^*(\mathfrak{g})$ of cochains on the Lie algebra $\mathfrak{g} = \text{Diff}_{\epsilon_2} \mathbb{C} \otimes \mathfrak{gl}_K$. A Koszul dual algebra of the Lie algebra cochain $C^*(\mathfrak{g})$ is the universal enveloping algebra $U(\mathfrak{g})$.

When $\epsilon_1 \neq 0$, $\text{Obs}^{5d}_{0,\epsilon_2}$ receives quantum corrections and deforms into an $A_\infty$ algebra[3] $\text{Obs}^{5d}_{\epsilon_1,\epsilon_2}$ and we denote the Koszul dual of it as $U_{\epsilon_1}(\mathfrak{g})$. Costello showed that $U(\mathfrak{g})$ has a non-trivial(section 9 of [17]) and unique(section 15 of [17]) deformation $U_{\epsilon_1}(\mathfrak{g})$ and identified it with the algebra of operators on $N$ M2 branes in the large $N$ limit.

As long as we do not break the supersymmetry and respect the topological holomorphic twist of the twisted supergravity, we can introduce M2 and M5 branes that inherit all twists in the background. They span the following directions in the twisted M-theory:

$$\begin{aligned} &\text{M2: } \mathbb{R}_t \times \mathbb{C}_{\epsilon_1}, \\ &\text{M5: } \{0\} \times \text{TN}_K \times \mathbb{C}_z. \end{aligned} \tag{7}$$

---

[3]For a physical description of $A_\infty$ algebra, see [29].

Each of the membranes supports an interesting quantum field theory [16, 17], which is a twisted subsector of 3d $\mathcal{N} = 4$ ADHM gauge theory and 6d $\mathcal{N} = (2, 0)$ SCFT, respectively. Due to the Omega background applied to each of the worldvolume, the field theories that encode the protected subsector are localized to $\mathbb{R}_t$ and $\mathbb{C}_z$:

$$\begin{aligned} &\text{M2: Topological quantum mechanics on } \mathbb{R}_t, \\ &\text{M5: Free fermion vertex operator algebra on } \mathbb{C}_z. \end{aligned} \tag{8}$$

In the limit of a large number of membranes, the algebra of operators on the M2 branes is 1-shifted affine $gl(1)$ Yangian with one central element eliminated and the algebra of operators on the M5 branes is $W_\infty$ algebra, or affine $gl(1)$ Yangian.

Note that in some literature, $\mathbb{C}_{\epsilon_1} \times \text{TN}_K$ part of the twisted M-theory background is replaced with

$$\mathbb{C}_{\epsilon_1} \times \frac{\mathbb{C}_{\epsilon_2} \times \mathbb{C}_{\epsilon_3}}{\mathbb{Z}_K}, \quad \text{where } \epsilon_1 + \epsilon_2 + \epsilon_3 = 0. \tag{9}$$

This presentation is useful to explain a triality automorphism of the algebra associated with the M2 and M5 branes(see [29], for instance). However, we will follow the original presentation [16], which did not show $\epsilon_2$, $\epsilon_3$ subscripts explicitly, since the triality does not play an important role in our discussion.

## 2.2 Compatibility of M2 and M5 branes in twisted M-theory

Let us comment on the compatibility of the M2 and M5 branes and the 11d topological holomorphic background and explain why these two BPS defects cannot engineer the domain wall-like defect, which is the main object of this paper.

M2 branes are entirely embedded in $\mathcal{M}_7^T$. It preserves 3d $\mathcal{N} = 4$ supersymmetry in the presence of the Taub-NUT geometry. There are two types of topological twists available in 3d: Rozansky-Witten twist [41] and the mirror version of it. Both twists are compatible with the twisted supergravity background and it is useful to work in either of the two twists to give a complete picture of the algebra of operators on the M2 branes. Clearly, the three-dimensional M2 branes cannot engineer the four-dimensional domain wall.

Next, out of the 6 directions of M5 brane worldvolume, 4 directions lie in $\mathcal{M}_7^T$ and 2 directions are in $\mathcal{M}_4^H$. There are two possible twists available to the 6d $(2, 0)$ theory.[4] First, a topological holomorphic twist that gives 4 topological directions and 2 holomorphic directions. Second, a holomorphic twist that gives all 6 directions holomorphic. Since we only have 4 holomorphic directions in the twisted M-theory background, only the first option is feasible and this background exactly applies to the M5 branes that appear in (8). In other words, since the 6d $(2, 0)$ theory does not admit a twist generating 2 topological and 4 holomorphic directions, we are not allowed to use M5 branes to engineer the domain wall-like defects in the 5d Chern-Simons theory, which require the M5 branes to wrap $\mathcal{M}_4^H$.

There are remaining BPS objects in M-theory: the M9 brane and the KK monopole. The KK monopole wraps different directions to those of the already existing KK monopole, which is equivalent to the Taub-NUT geometry.

## 2.3 KK monopole, intersecting D6 branes and a domain wall

We would like to engineer a codimension-1 defect in the 5d Chern-Simons theory from the remaining BPS objects. However, the M9 brane only admits a holomorphic twist (on the entire

---

[4]We are grateful to Kevin Costello for the discussion on this point.

10 directions) on its worldvolume, so we exclude the M9 brane since it is not compatible with the 7 topological directions. Next, let us consider

$$\text{New KK monopole: } \mathbb{C}_z \times \mathbb{C}_w \times \mathbb{R}^3, \tag{10}$$

where $\mathbb{R}^3 \subset \mathcal{M}_7^T$ (we will specify the $\mathbb{R}^3$ more precisely later in the IIA frame.) Since the 7d SYM on the worldvolume of the KK monopole admits a topological holomorphic twist that induces 3 topological and 4 holomorphic directions, like the original KK monopole associated with $TN_K$, the new KK monopole is compatible with the twisted background.

Let us now argue that the new KK monopole is compatible with the other necessary conditions for twisted M-theory: 1. $G_2$-holonomy, 2. $S_{\epsilon_1}^1 \times S_{\epsilon_2}^1$ action.

First, the multi(original plus new) KK monopole configuration corresponds to a singular $G_2$-holonomy manifold [37, 38], if we tune $\mathbb{R}^3$ to intersect with $\mathbb{R}_t \times \mathbb{C}_{\epsilon_1}$ at one point. If there are $N$ new KK monopoles, the singularity of the $G_2$-holonomy manifold represents the unfolding of an $A_{N+K}$ singularity into $A_N$ and $A_K$ singularities [42]. More precisely, it is a cone on a weighted projective space $\mathbf{WCP}_{N,N,K,K}^3$ [37,38]. Let us call this new geometry $\widetilde{\mathcal{M}}_7^T$.

Second, we will argue that $\widetilde{\mathcal{M}}_7^T$ admits $S_{\epsilon_1}^1 \times S_{\epsilon_2}^1$ isometry. The original Taub-NUT circle $S_{TN}^1$ is intact by the introduction of new KK monopoles. $\widetilde{\mathcal{M}}_7^T$ has an $S_{\epsilon_2}^1$ isometry that rotates $S_{TN}^1$. Indeed, when we go to the type IIA frame, we take this circle as the M-theory circle to reduce.[5] Next, it is helpful to go to the type IIA frame to see the presence of $S_{\epsilon_1}^1$ action. Reducing along $S_{TN}^1$, we map two types of KK monopoles in the M-theory to $K$ D6 and $N$ D6' branes in the type IIA frame that wrap the 4 holomorphic directions and 3 topological directions:

$$\begin{aligned} &\text{D6: } \mathbb{C}_z \times \mathbb{C}_w \times \mathbb{R}_t \times \mathbb{C}_{\epsilon_1}, \\ &\text{D6': } \mathbb{C}_z \times \mathbb{C}_w \times \mathbb{R}^3. \end{aligned} \tag{11}$$

In general, such a D6-D6' configuration does not preserve supersymmetry. However, if we align them to make certain angles, we can preserve 4d $\mathcal{N} = 1$ supersymmetry on $\mathbb{C}_z \times \mathbb{C}_w$ [39]. More precisely, we may split the 6 topological directions($\widetilde{\mathcal{M}}_7^T / S_{TN}^1$) into three 2-planes $\mathcal{P}_t$, $\mathcal{P}_1$, $\mathcal{P}_2$. Then, we embed each $\mathbb{R}_i$ of $\mathbb{R}_t \times \mathbb{C}_{\epsilon_1} = \mathbb{R}_t \times \mathbb{R}_1 \times \mathbb{R}_2$ of D6 branes in $\mathcal{P}_i$, respectively. We parametrize $\mathbb{R}_i$ with $x_i$ and the orthogonal axis in $\mathcal{P}_i$ with $y_i$. Next, we orient each $\mathbb{R}_i'$ of $\mathbb{R}^3 = \mathbb{R}_t' \times \mathbb{R}_1' \times \mathbb{R}_2'$ of the D6' branes in $\mathcal{P}_i$ to make an angle $\varphi_i$ with $\mathbb{R}_i$ such that [39]

$$\varphi_t \pm \varphi_1 \pm \varphi_2 = 0 \mod 2\pi \tag{12}$$

for some choice of signs.

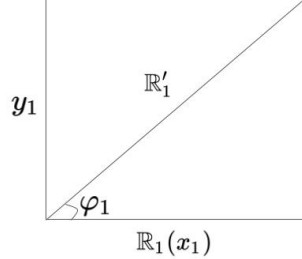 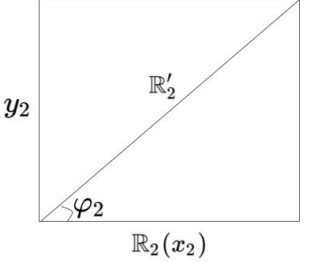

Figure 1: $\mathcal{P}_1$ and $\mathcal{P}_2$. Each of them is parametrized by $(x_1, y_1)$ and $(x_2, y_2)$ respectively.

---

[5]Historically, the brane configuration, which we get after the circle reduction in type IIA, was first discovered [39] and M-theory lift was introduced later [37, 38].

Given the supersymmetry, let us check the $S^1_{\epsilon_1}$ isometry in $\widetilde{\mathcal{M}}^T_7/S^1 \sim \mathbb{R}_t \times \mathbb{C}_{\epsilon_1} \times \mathbb{R}^3_{TNB}$. Recall that we took $TN_K \approx S^1_{TN} \times \mathbb{R}^3_{TNB}$. In the present case, it suffices to show that the metric on both D6 and D6' branes is invariant under $S^1_{\epsilon_1}$ action that simultaneously rotates $\mathbb{C}_{\epsilon_1} \times \mathbb{R}^2_{tnb}$, where $\mathbb{R}^2_{tnb}$ is a 2-plane embedded in $\mathbb{R}^3_{TNB}$.

Let us first consider two independent rotations, parametrized by $\theta_1$, $\theta_2$, respectively. They act on the coordinates $(x_1, x_2)$ of $\mathbb{C}_{\epsilon_1}$ and $(y_1, y_2)$ of $\mathbb{R}^2_{tnb}$ as follows

$$\begin{pmatrix} x_1 \\ x_2 \end{pmatrix} \to \begin{pmatrix} \cos\theta_1 x_1 - \sin\theta_1 x_2 \\ \sin\theta_1 x_1 + \cos\theta_1 x_2 \end{pmatrix}, \quad \begin{pmatrix} y_1 \\ y_2 \end{pmatrix} \to \begin{pmatrix} \cos\theta_2 y_1 - \sin\theta_2 y_2 \\ \sin\theta_2 y_1 + \cos\theta_2 y_2 \end{pmatrix}. \tag{13}$$

The relevant part of the metric on the original D6 branes is preserved trivially under the rotation:

$$ds^2 = dx_1^2 + dx_2^2 \to dx_1^2 + dx_2^2. \tag{14}$$

If we denote $a = \tan\varphi_1$, $b = \tan\varphi_2$, we may write down the relevant part of the metric on the new D6' branes as

$$\begin{aligned} ds^2 &= d(x_1 + ay_1)d(x_1 + ay_1) + d(x_2 + by_2)d(x_2 + by_2) \\ &= dx_1^2 + dx_2^2 + a^2 dy_1^2 + b^2 dy_2^2 + a(dx_1 dy_1 + dy_1 dx_1) + b(dx_2 dy_2 + dy_2 dx_2). \end{aligned} \tag{15}$$

For (15) to be invariant under (13), $a$, $b$, $\theta_1$, $\theta_2$ need to satisfy the following set of equations:

$$\begin{aligned} a^2 \sin^2\theta_2 + b^2 \cos^2\theta_2 &= b^2, \\ a^2 \cos^2\theta_2 + b^2 \sin^2\theta_2 &= a^2, \\ (a^2 - b^2)\cos\theta_2 \sin\theta_2 &= 0, \\ a\cos\theta_1 \cos\theta_2 + b\sin\theta_1 \sin\theta_2 &= a, \\ a\sin\theta_1 \sin\theta_2 + b\cos\theta_1 \cos\theta_2 &= b, \\ a\cos\theta_1 \sin\theta_2 - b\sin\theta_1 \cos\theta_2 &= 0, \\ a\sin\theta_1 \cos\theta_2 - b\cos\theta_1 \sin\theta_2 &= 0. \end{aligned} \tag{16}$$

There is an obvious solution:

$$a = b \quad \text{and} \quad \theta_1 = \theta_2. \tag{17}$$

In principle, $a = b$ can be anything, but to be consistent with Figure 4, let us choose them to be 1. As this condition fixes $\varphi_1 = \varphi_2 = \pi/4$, by (12), we can also fix $\varphi_t$. Since we want to preserve exactly $\mathcal{N} = 1$ supersymmetry on the domain wall, $\varphi_t$ should not be 0,[6] but any value, which satisfies (12). The existence of the solution (17) guarantees the $U(1)_{\epsilon_1}$ isometry, which simultaneously($\theta_1 = \theta_2$) rotates $\mathbb{C}_{\epsilon_1} \times \mathbb{R}^2_{tnb}$.

Now that we have shown the compatibility of the new KK monopoles with the twisted M-theory, let us discuss the degree of freedom that they introduce. It is helpful to stay in the type IIA frame to do so. There are D6-D6' strings localized at the 4d intersection $\mathbb{C}_z \times \mathbb{C}_w$. The low energy mode of the quantization of D6-D6' strings corresponds to the 4d $\mathcal{N} = 1$ massless chiral multiplet [39], charged under both Chan-Paton symmetries associated with the D6 and D6' branes. We can also understand this matter content as off-diagonal components in the decomposition of $A_{N+K}$ singularity into $A_N$ and $A_K$ singularities in the M-theory [37, 38, 42]. Since the worldvolume theory of D-branes probing the twisted supergravity background inherits the twist, the D6-D6' string or the 4d $\mathcal{N} = 1$ chiral multiplet on $\mathcal{M}^H_4 = \mathbb{C}_z \times \mathbb{C}_w$ is holomorphically twisted.

---

[6]In this case, the D6 and D6' branes intersect in 5 directions, which is not of our interest. In fact, configuration with $\varphi_t = 0$ and its M-theory uplift was discussed in [37, 38], but another set of KK monopoles other than the two sets of KK monopoles should be considered in that case.

Finally, as we have seen in §2.1, because of the Omega background on the 2-plane $\mathbb{R}^2_{tnb}$ inside the $N$ D6' branes, the theory on the $N$ D6' branes localizes on another copy of the 5d $U(N)$ CS theory. However, this theory will not play an important role in the following discussion, as we will focus on the 4d intersection.

# 3 4d VOA in the 5d Chern-Simons theory

To construct a set of physical operators on the domain wall, we need to collect all fields, which live on the domain wall. In §3.1, we will describe the first ingredient, the holomorphically twisted 4d $\mathcal{N}=1$ chiral multiplet coming from the D6-D6' strings. We can recast the 4d holomorphic field theory in a first-order formalism, called a twisted formalism in [44], and find that the 4d Lagrangian resembles that of the 2d $\beta\gamma$ system. The authors of [14] derived the 4d $\beta\gamma$ system. In §3.1, §3.2, we will translate and explain this mathematical notion in the more familiar language to physicists using holomorphic descent, which is an analog of the topological descent defined in Witten's classical paper [1]. This is in the line of [43–45], where the authors defined the topological holomorphic descent and the product between a local operator and a descent operator, which is called a secondary product. In particular, see the beautiful exposition in the paper [45] by Beem, Ben-Zvi, Bullimore, Dimofte, and Neitzke. In §3.3, we identify the second ingredient: the boundary condition of the 5d Chern-Simons gauge field on the domain wall. Combining these ingredients, we construct the physical operators living on the domain wall in §3.4. The 4d $\beta\gamma$ system on the domain wall, however, is anomalous. The anomaly cancelation condition is a shift of the non-commutativity parameter $\epsilon_2$ on one side of the domain wall, compared to that on the other side, as we show in §3.5.

## 3.1 Holomorphic twist of 4d $\mathcal{N}=1$ theory

Let us study the D6-D6' strings that are localized on the D6, D6' intersection $\mathbb{C}_z \times \mathbb{C}_w$. The D6-D6' strings give the 4d $\mathcal{N}=1$ chiral multiplet $\Phi = (\phi, \psi_\alpha)$ and the D6'-D6 strings give the complex conjugate $\bar{\Phi} = (\bar{\phi}, \bar{\psi}_{\dot{\alpha}})$, where $\alpha$ and $\dot{\alpha}$ are spinor indices of $SU(2)_l \times SU(2)_r = SO(4)_{Lorentz}$. Since branes moving in the twisted supergravity background inherit the twist of the background, we get the holomorphically twisted 4d $\mathcal{N}=1$ chiral multiplet theory. We will define and study the holomorphic twist of 4d $\mathcal{N}=1$ theory in this subsection.

4d $\mathcal{N}=1$ supersymmetry algebra is generated by four supercharges $Q_\alpha$ and $\bar{Q}_{\dot{\alpha}}$:

$$
\begin{aligned}
[Q_\alpha, Q_\beta] &= [Q_{\dot{\alpha}}, Q_{\dot{\beta}}] = 0, \\
[Q_\alpha, Q_{\dot{\beta}}] &= \sigma^\mu_{\alpha\dot{\beta}} P_\mu,
\end{aligned}
\tag{18}
$$

where $\mu$ is the 4d spacetime index, $\sigma^0 = -\text{Id}$, and $\sigma^{1,2,3}$ are Pauli matrices. The graded commutator[7] $[\ ,\ ]$ is defined by $[a, b] = ab - (-1)^{F(a)F(b)} ba$, where $F(a)$ is a fermion number of $a$.

Under the holomorphic twist, the Lorentz symmetry algebra is redefined in such a way that the supersymmetry generators are reorganized into a scalar $Q$ and a 1-form $\mathbf{Q}$ under the new Lorentz symmetry algebra. They satisfy the following relations:

$$
\begin{aligned}
Q^2 &= 0, \\
[Q, \mathbf{Q}] &= iP_{\bar{z}} d\bar{z} + iP_{\bar{w}} d\bar{w}.
\end{aligned}
\tag{19}
$$

---

[7]We unify the convention for the commutator to distinguish it from the secondary product $\{\ ,\ \}$, which we will soon introduce.

We may solve (19) utilizing (18) and find the scalar supercharge

$$Q = Q_-, \tag{20}$$

which defines the Q-cohomology and the 1-form supercharge **Q**, which anti-commutes with $Q$ to produce anti-holomorphic translation generators $P_{\bar{z}}$, $P_{\bar{w}}$:

$$\mathbf{Q} = \bar{Q}_{\dot{+}} d\bar{z} - \bar{Q}_{\dot{-}} d\bar{w}. \tag{21}$$

Given a local operator $\mathcal{O}$, one may act with **Q** n-times to obtain an n-form operator $\mathcal{O}^{(n)}$

$$\mathbf{Q}^n \mathcal{O} = \mathcal{O}^{(n)}. \tag{22}$$

There are two kinds of operators in the Q-cohomology. The first type is a local operator $\mathcal{O}(z)$. Examples of the first type operator in 4d $\mathcal{N} = 1$ chiral multiplet theory are

$$\bar{\phi}, \quad \psi_+, \tag{23}$$

including all of their holomorphic derivatives with respect to $\partial_z$ and $\partial_w$. See Appendix A for the derivation. Even though $\psi_+$ now transforms as $(2,0)$ form under the new Lorentz symmetry of the twisted theory, we will keep the notation and not explicitly show the differential form in the later discussion for a concise presentation. We will also do so for the other components of the spinor $\psi_-$ and $\bar{\psi}_{\dot{\alpha}}$, which are a scalar and $(0,1)$-form under the new Lorentz symmetry.

The product of two first-type operators

$$\mathcal{O}_{1,2}(z_1, z_2) = \mathcal{O}_1(z_1)\mathcal{O}_2(z_2) \tag{24}$$

is again in the Q-cohomology. By Hartog's theorem, in $d \geq 2$-complex dimension, every function on $U \subset \mathbb{C}^d$ that is holomorphic away from singularities can be uniquely extended to a holomorphic function on the entire $U$. Therefore, there cannot be a singular OPE between two local operators. This is in contrast with a complex one-dimension holomorphic theory, which is the usual vertex operator algebra, where one obtains interesting mode algebras of various holomorphic currents due to the singular OPE's between the currents.

The second type of operator in the Q-cohomology is more interesting. We can construct it by first applying the 1-form supercharge to the first-type operator. The resulting $\mathcal{O}^{(1)}$ is not Q-closed, but we can check

$$Q\mathcal{O}^{(1)} = \bar{\partial}\mathcal{O}. \tag{25}$$

If we then wedge $\mathcal{O}$ with the holomorphic top form $\Omega$ of $\mathbb{C}^2$, we get

$$Q(\Omega \mathcal{O}^{(1)}) = d(\Omega \mathcal{O}). \tag{26}$$

By integrating both sides along $S^3$, which does not have a boundary, we get a Q-closed operator at $x$

$$Q\left[\int_{S_x^3} \Omega \wedge \mathcal{O}^{(1)}\right] = 0, \tag{27}$$

where $S_x^3$ is a 3-sphere centered at $x$. It may look nonlocal, as it involves the integral over $S^3$, but it is a local operator at $x$ after the integral. We can avoid a potential intersection with other second-type operators centered at different points by shrinking $S^3$'s. For two second-type operators sharing the same center, there may be a nontrivial algebraic structure. We will describe more the algebra in the next subsection.

Examples of the second-type operator in 4d $\mathcal{N} = 1$ chiral multiplet theory are

$$
\begin{aligned}
\int_{S^3} dzdw\, \bar{\phi}^{(1)} &= \int_{S^3} dzdw\, (\bar{\psi}_+ d\bar{z} - \bar{\psi}_- d\bar{w}), \\
\int_{S^3} dzdw\, \psi_+^{(1)} &= \int_{S^3} dzdw\, (\partial_w \phi\, d\bar{z} - \partial_z \phi\, d\bar{w}).
\end{aligned}
\tag{28}
$$

We have collected the supersymmetry transformations of the components of the 4d $\mathcal{N} = 1$ chiral multiplet under $Q$ and $\mathbf{Q}$ in Appendix A.

Furthermore, we can define a secondary product between the first-type operator and the second-type operator by wrapping the first-type operator with a 3-sphere that is used to define the second-type operator

$$
\{\mathcal{O}_1, \mathcal{O}_2\}(z_0, w_0) = \int_{S^3_{z_0,w_0}(z,w)} \mathcal{O}_1(z_0, w_0)\, \mathcal{O}_2^{(1)}(z,w) dz dw,
\tag{29}
$$

where $S^3_{z_0,w_0}(z,w)$ is a 3-sphere centered at $(z_0, w_0)$ and parametrized by two complex coordinates $z$, $w$. Here $\mathcal{O}_1$ and $\mathcal{O}_2$ are cochain representatives and the secondary product does not depend on the choice of the representatives, since Q-exact deformation of $\mathcal{O}_i$ will not change the result.

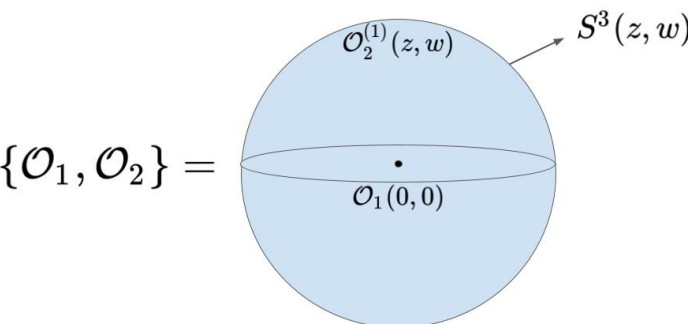

Figure 2: It illustrates a product between the first and second-type operators. We place $\mathcal{O}_1$ at the origin and wrap it with $\Omega\mathcal{O}_2^{(1)}$, which is supported on the 3-sphere $S^3$, parametrized by $z, w$.

Again, $\{\mathcal{O}_1, \mathcal{O}_2\}(z_0, w_0)$ is a local operator in the Q-cohomology, supported at $(z_0, w_0)$. To compute the secondary product, we need to make a contraction between $\mathcal{O}_1$ and $\mathcal{O}_2^{(1)}$. Hence, the result of the secondary product is in general a first-type operator. It is the remaining operator, which is not contracted. More details of the secondary product along with the study of the holomorphic twist of a general 4d $\mathcal{N} = 1$ gauge theory will be discussed in [51].

It is helpful to write down the Lagrangian for the 4d $\mathcal{N} = 1$ chiral multiplet to extract the propagators used in the contraction:

$$
\int_{\mathbb{C}_z \times \mathbb{C}_w} \partial\phi\, \bar{\partial}\bar{\phi} + \psi_+ \partial_{\bar{z}}\bar{\psi}_- + \psi_+ \partial_{\bar{w}}\bar{\psi}_+ + \dots
\tag{30}
$$

From the kinetic terms, we read off the nonvanishing propagators that involve the first-type operators in the Q-cohomology:

$$
\langle \bar{\phi}\, \partial\phi \rangle \sim \langle \psi_+ \bar{\psi} \rangle \sim \frac{\omega_{BM}}{dzdw},
\tag{31}
$$

where $\omega_{BM}$ is Bochner-Martinelli kernel

$$\omega_{BM} = dz\,dw\,\frac{\bar{z}d\bar{w} - \bar{w}d\bar{z}}{|z|^2 + |w|^2}. \tag{32}$$

The Bochner-Martinelli kernel is a four-dimensional analog of the $S^1$ residue integral measure $\frac{dz}{z}$, used in 2d VOA mode algebra computation. It has the following nice property that will play an important role in the next subsection:

$$\int_{S^3_{z_0,w_0}} \omega_{BM} f(z,w) = f(z_0, w_0). \tag{33}$$

## 3.2 Higher vertex operator algebra

Before we discuss the 4d VOA, let us recall the familiar example of the OPE and residue pairing in a simple 2d VOA; we will find the analogous properties in the 4d soon. Consider 2d $\beta\gamma$ system, which has the following OPE:

$$\beta(z) \cdot \gamma(0) \sim \frac{1}{z} + \dots, \tag{34}$$

where $\dots$ are regular terms. Then, we may compute the following integral

$$\int_{S^1_0} dz(\beta(z) \cdot \gamma(0)) = \int_{S^1_0} \frac{dz}{z} = 1, \tag{35}$$

by using the familiar residue formula.

Similar to 2d, in 4d VOA we form an "OPE" between the first-type and second-type operators

$$\mathcal{O}_1(z_0, w_0) \cdot \mathcal{O}_2^{(1)}(z, w). \tag{36}$$

Note that a more general OPE in holomorphically twisted field theory was discussed in [11].

Next, we can compute the $S^3$ residue integral, using the nontrivial propagators defined in (31):

$$\int_{S^3_{z_0,w_0}} dz\,dw\; \mathcal{O}_1(z_0, w_0) \cdot \mathcal{O}_2^{(1)}(z, w). \tag{37}$$

This is precisely the secondary product $\{\mathcal{O}_1, \mathcal{O}_2\}$, which we defined in (29).

Specializing to the 4d $\mathcal{N} = 1$ theory of a chiral multiplet, we can compute a simple secondary product:

$$\{\bar{\phi}, \psi_+\} = \int_{S^3_0} dz \wedge dw \wedge \bar{\phi} \cdot \psi_+^{(1)} = \int_{S^3_0} \omega_{BM} = 1. \tag{38}$$

We will soon analyze a product involving two second-type operators as well.

We can make the analogy more concrete by rewriting the Lagrangian in the descent field utilizing the BV Lagrangian [14]. The physical Lagrangian (30) admits a compact representation in terms of the following twisted superfields

$$\begin{aligned} \gamma &= (\bar{\phi}, \bar{\phi}^{(1)}, \bar{\phi}^{(2)}), \\ \beta &= (\psi_+, \psi_+^{(1)}, \psi_+^{(2)}). \end{aligned} \tag{39}$$

By multiplying $\beta$, $\gamma$ so that the form degree of the integrand sums up to $(2, 2)$, we can rewrite (30) as a part of the bigger BV Lagrangian that follows

$$\int_{\mathbb{C}_z \times \mathbb{C}_w} dz dw \beta \bar{\partial} \gamma. \tag{40}$$

This free field Lagrangian resembles 2d VOA $\beta\gamma$ system. As our main intention to present this expression is to show the resemblance of the higher VOA to the usual 2d VOA, we will not try to give details on the BV formalism. The reader who is interested in this may refer to [14,44].

Let us now define a higher current that generates a higher VOA, which is an analog of the affine current operator in the 2d VOA. Rather than giving an abstract definition, we will explain using the 4d $\mathcal{N} = 1$ chiral multiplet example, as it will be our main interest. We define a *precurrent J*

$$J = \bar{\phi}\psi_+ \tag{41}$$

and consider its first descendant $J^{(1)}$:

$$J^{(1)} = (-\bar{\psi}_{\dot{+}} d\bar{w} + \bar{\psi}_{\dot{-}} d\bar{z})\psi_+ + \bar{\phi}(\partial_w \phi \, d\bar{z} - \partial_z \phi \, d\bar{w}). \tag{42}$$

This is not in Q-cohomology by the general discussion above, but we can obtain a nontrivial Q-cohomology class by combining it with the holomorphic top form on $\mathbb{C}^2$ and integrating the wedged product over $S^3$:

$$\int_{S^3} dz \wedge dw \wedge J^{(1)}. \tag{43}$$

We call $dz \wedge dw \wedge J^{(1)}$ a *higher current*. Notice that $J^{(1)}$ captures some components of the flavor current operator in the 4d $\mathcal{N} = 1$ theory of a free chiral multiplet. In this sense, the expression (43) is a charge $Q$ of a conserved global symmetry current $j$, which is defined as

$$Q = \int_{S^{2n-1}} \star j \tag{44}$$

in the $2n-$dimensional spacetime, where $\star$ is the Hodge star of the spacetime geometry. The higher current can be compared with $\star j$ in (43). Indeed, it satisfies the conservation law up to a Q-exact term,[8] which we can treat as zero:

$$d(dz dw J^{(1)}) = Q(dz dw J^{(1)}) \equiv 0. \tag{45}$$

In principle, one can think about the multiplication operation by a monomial $z^m w^n$ in the integrand of (43) and obtain a double graded associative algebra of modes,[9] which is a generalization of the single-graded algebra of modes of the 2d VOA. Therefore, we can think (43) as a zero mode($m = n = 0$) of the higher current. This observation will be useful when we construct the algebra of operators on the domain wall by combining all ingredients.

We have studied the holomorphically twisted 4d $\mathcal{N} = 1$ chiral multiplet, which comes from the D6-D6' string. Since the domain wall separates the 5d Chern-Simons theory, we also need to take care of the boundary condition of the 5d gauge field on the domain wall to construct the algebra of operators on the domain wall. It will be the second ingredient.

---

[8]See also the relevant discussion in Section 3.5 of [43]. We thank Junya Yagi for useful discussion on this point.

[9]One should distinguish a higher VOA from a toroidal Lie algebra [46], since the former is related to $(\mathbb{C}^2)^\times \sim S^3$ rather than $(\mathbb{C}^\times)^2 \sim T^2$ by construction (43). Indeed, the original reference of the higher VOA [12] called it the sphere algebra.

### 3.3 The boundary condition of the 5d gauge field

Before considering the boundary condition of the 5d gauge field $A_{5d}$, we first notice that we can interpret the boundary condition of the 5d gauge field as the background gauge field for the flavor symmetry $U(K)$ of the 4d $\beta\gamma$ system. The induced non-dynamical coupling modifies the free kinetic term of the 4d $\beta\gamma$ system in the following way,

$$\int_{\mathbb{C}_z \times \mathbb{C}_w} dz dw \wedge \beta \left( \bar{\partial} + A_{5d}|_\partial \right) \gamma. \tag{46}$$

Here, the anti-holomorphic derivative is replaced by the gauge covariant derivative with respect to the 5d gauge field $A_{5d}$ at the boundary $t = 0$, $A_{5d}|_\partial$. The form of this minimal interaction indicates that the 5d gauge field $A_{5d} = A_t dt + A_{\bar{z}} d\bar{z} + A_{\bar{w}} d\bar{w}$ loses its t-component $A_t$ when it approaches to the boundary.

Next, we will follow the standard approach used to study the boundary condition of the 3d Chern-Simons theory. We will turn off the non-commutativity parameter temporarily as it does not affect the following analysis. We need to make sure of the locality of the 5d Chern-Simons theory near the boundary $t = 0$. To show that, let us vary the action, obtaining

$$\delta S_{5d\ CS} = \int_{\mathbb{R}_t \times \mathbb{C}_z \times \mathbb{C}_w} dz \wedge dw \wedge \mathrm{Tr} \left( \delta A \wedge dA + d(\delta A \wedge A) \right). \tag{47}$$

Using Stokes theorem, we may rewrite the second term as

$$\int_{\mathbb{C}_z \times \mathbb{C}_w} dz \wedge dw \wedge \delta A \wedge A. \tag{48}$$

From this, we see two types of equivalently allowed boundary conditions

$$A_{\bar{z}} = 0 \quad \text{or} \quad A_{\bar{w}} = 0, \tag{49}$$

which makes the boundary variations (48) zero. This resembles the familiar holomorphic Dirichlet boundary condition of the 3d Chern-Simons theory. We will choose to work with the second one $A_{\bar{w}} = 0$.

By the equation of motion of the 5d Chern-Simons theory,

$$(d_t + \bar{\partial}_{\mathbb{C}_z \times \mathbb{C}_w}) A_{5d} = 0, \tag{50}$$

and the boundary condition that we have just fixed, we conclude the boundary condition of the 5d gauge field at the domain wall is a holomorphic function in $z$ and $w$, valued in the Lie algebra $\mathfrak{gl}_K$:

$$A_{5d}|_\partial = A_{\bar{z}}(z, m) \in \mathbb{C}[z, w] \otimes \mathfrak{gl}_K. \tag{51}$$

### 3.4 Physical observables on the domain wall

So far, we have collected the ingredients that live at the domain wall in the 5d Chern-Simons theory. Those are the holomorphic twist of 4d $\mathcal{N} = 1$ chiral multiplet and the boundary condition of the 5d Chern-Simons theory. We will combine those degrees of freedom and construct the gauge invariant physical operators living on the domain wall. The relevant gauge symmetry is $U(N)$, where $N$ is the number of D6' branes. On the other hand, we treat $U(K)$ as a flavor symmetry,[10] where $K$ is the number of D6 branes.

---

[10]Since both D6 and D6' branes are non-compact, one may think both $U(N)$ and $U(K)$ groups to be non-dynamical. However, D6 is related to the bulk dynamics and D6' is related to the boundary dynamics. Since we are now studying the boundary dynamics on D6', we may treat D6 heavy and only freeze the associated gauge field. [16, 40] also discussed such an assignment of the flavor and gauge symmetry in an analogous intersecting D4-D6 brane system.

We have already seen a candidate for the gauge invariant algebra of operators in §3.2

$$\mathcal{C}[1] = \int_{S^3_0(z,w)} dz \wedge dw \wedge J^{(1)}. \tag{52}$$

We will shortly clarify the notation used in the LHS of (52). Restoring all indices of the pre-current J, we have

$$J^i_j(z,w) = \bar{\phi}^i_a \psi^a_{+j}, \tag{53}$$

where $a$ is the $U(N)$ gauge symmetry and $i, j$ are the $U(K)$ flavor symmetry index. Note that we treat $J^i_j(z,w)$ as a composite operator, which is a holomorphic function; there is no UV divergence in forming such an operator out of $\bar{\phi}$ and $\psi_+$, since there is no non-trivial OPE between a boson and a fermion. As we have remarked in §3.2, we can multiply a holomorphic function to the integrand of (52), the result is again well-defined. Now, recall from §3.3 that the boundary condition of the 5d gauge field is precisely a holomorphic function in $z$ and $w$, valued in the Lie algebra $\mathfrak{gl}_K$. Therefore, we can combine (52) and

$$A_{5d}|_\partial \in \mathbb{C}[z,w] \otimes \mathfrak{gl}_K \tag{54}$$

to form a generating set of the algebra of operators on the domain wall.

Consequently, we propose that the generators of the algebra of operators on the domain wall are

$$\mathcal{C}[z^m w^n T^A] = \int_{S^3_0(z,w)} dz dw\, z^m w^n T^A\, J^{(1)}. \tag{55}$$

$m, n$ are non-negative integers and $T^A \in \mathfrak{gl}_K$, where $A \in \{1, \dots, K^2\}$. A few remarks are in order.

First, if there is a 4d gauge field localized at the domain wall, we need to introduce $b$, $c$ ghosts that would modify the precurrent $J$ and the higher current $J^{(1)}$, as well. However, the gauge field is a 7d field that lives in the 7d worldvolume of the $N$ D6' branes. Unless we compactify the three transverse directions normal to the domain wall, this cannot be treated as the 4d gauge field. Hence, we do not have to introduce the ghosts for the 7d gauge field. Moreover, we do not introduce ghosts for constant gauge transformations. Therefore, all we need to do is to take a trace in $\mathfrak{gl}_N$, which we have already done in (53).[11] Second, notice that (55) is not a secondary product, defined in (29), between $\mathcal{C}[1]$ and $A_{5d}|_\partial$. Even though the secondary product involves a second-type operator, the result of the product is a first-type operator, which is not expected to generate a nontrivial algebra. We will see why such an operator does not form a nontrivial algebra in §4.

This generator is a four-dimensional version of the two-dimensional observable considered in Proposition 15.3.1 of [16], which discussed the twisted holography of $N$ M5 branes,

$$\mathcal{C}_{2d}\left[A_{5d}|_{\mathbb{C}_z} = z^m \partial_z^n T^A\right] = \int_{S^1_0(z)} dz\, z^m \partial_z^n T^A\, J. \tag{56}$$

In this case, the stack of M5 branes forms a surface defect supported on $\mathbb{C}_z$ in the 5d CS theory. $A_{5d}|_{\mathbb{C}_z}$ is a $\bar{z}$ component of the 5d gauge field on $\mathbb{C}_z$. $J$ is a fermion bilinear $\psi\bar{\psi}$ and it is a local operator, which does not involve any descent. The pair of fermions is a quantization of D6-D4 strings. The D4 branes, which are avatars of the M5 branes in type IIA frame, intersect with the D6 branes along $\mathbb{C}_z$. The author of [16] proved the isomorphism between the algebra of operators on $N$ D4 branes, generated by (56), and the $W_{K+\infty}$ algebra in the large $N$ limit. Note that we may also write $w$ in (55) as $\partial_z$, due to the non-commutative background on $\mathbb{C}_z \times \mathbb{C}_w$, which induces a non-trivial commutator between $z$ and $w$:

$$[z,w] = \epsilon_2 \quad \text{or} \quad [z,\partial_z] = \epsilon_2. \tag{57}$$

---

[11]We are thankful to Kevin Costello, who patiently explained the relevant part of his paper [16].

### 3.5 5d/4d system and anomaly cancellation

Before we close the section, we would like to point out a subtle modification of the 5d Chern-Simons theory on one side of the domain wall compared to another side of the wall, due to the anomaly coming from the domain wall. After the modification, the 5d/4d system is anomaly-free. Although we have treated the $U(K)$ gauge field as a background field when we studied the boundary dynamics, we still need to cancel the $U(K)$ gauge anomaly for the bulk 5d $U(K)$ gauge theory to make sense.

The Lagrangian of the 5d-4d coupled system is

$$\int_{\{0\}\times\mathbb{C}_z\times\mathbb{C}_w} \beta\bar{\partial}_A\gamma + \frac{1}{\epsilon_1}\int_{\mathbb{R}_t\times\mathbb{C}_z\times\mathbb{C}_w} dzdw\,\mathrm{Tr}\left(AdA + \frac{2}{3}A^3 + 2\epsilon_2 A\{A,A\}\right) + \dots \tag{58}$$

Recall that in the presence of the gauge field $A$, the partition function of the $\beta\gamma$ system is a local section of a determinant line bundle $\mathcal{L}_{\det}$ on the moduli space of connections of the base manifold (here is $\mathbb{C}^2_{zw}$). In order to quantize $A$, we need to have a well-defined partition function in the first place and the local anomaly is the curvature of the determinant line bundle $F_{\det}$, which is determined by the following formula [47, Theorem 10.35]:

$$\int_D F_{\det} = n\int_{\mathbb{C}_z\times\mathbb{C}_w\times D} \mathrm{Tr}(F_A^3), \tag{59}$$

where $D$ is an arbitrary 2 dimensional region in the moduli of connections, $n$ is some constant independent of $D$, and $F_A = dA + A\wedge A$ is the curvature form of $A$. To trivialize $\mathcal{L}_{\det}$, we start from the partition function at $A = 0$, find a path $L$ connecting 0 and $A$, and define the partition function at $A$ to be the parallel transport of $Z_{A=0}$ along $L$. If we perturb $L$ to $L'$ connecting 0 and $A$, then the difference between parallel transports along $L$ and $L'$ is $\int_D F_{\det}$, where $D$ is a disk bounded by $L$ and $L'$. Using Stokes formula, we can write it as

$$n\int_{L\times\mathbb{C}_z\times\mathbb{C}_w} \mathrm{CS}_{5d}(A) - n\int_{L'\times\mathbb{C}_z\times\mathbb{C}_w} \mathrm{CS}_{5d}(A), \tag{60}$$

where $\mathrm{CS}_{5d}(A)$ is the ordinary 5d Chern-Simons form

$$\mathrm{CS}_{5d}(A) = \mathrm{Tr}\left(A\wedge dA\wedge dA + \frac{3}{2}A^3\wedge dA + \frac{3}{5}A^5\right). \tag{61}$$

Since $A$ only has anti-holomorphic components along $\mathbb{C}_z\times\mathbb{C}_w$, we can rewrite $A\wedge dA\wedge dA$ as $A\wedge\partial A\wedge\partial A$, and all other parts of $\mathrm{CS}_{5d}(A)$ vanish, and we see that the action functional

$$\int_{\{0\}\times\mathbb{C}_z\times\mathbb{C}_w} \beta\bar{\partial}_A\gamma - n\int_{\mathbb{R}_{t\geq 0}\times\mathbb{C}_z\times\mathbb{C}_w} \mathrm{Tr}(A\wedge\partial A\wedge\partial A) \tag{62}$$

is free of gauge anomaly, where we put the boundary condition that $A\to 0$ when $t\to\infty$.

On the other hand, separating out the 5d holomorphic Chern-Simons action into two domains, we get

$$\int_{\{0\}\times\mathbb{C}_z\times\mathbb{C}_w} \beta\bar{\partial}\gamma + \frac{1}{\epsilon_1}\int_{\mathbb{R}_t^-\times\mathbb{C}_z\times\mathbb{C}_w} dzdw\,\mathrm{Tr}\left(AdA + \frac{2}{3}A^3 + 2\epsilon_2 A\{A,A\}\right)$$
$$+ \frac{1}{\epsilon_1}\int_{\mathbb{R}_t^+\times\mathbb{C}_z\times\mathbb{C}_w} dzdw\,\mathrm{Tr}\left(AdA + \frac{2}{3}A^3 + 2\epsilon_2' A\{A,A\}\right), \tag{63}$$

which can be rewritten as

$$\frac{1}{\epsilon_1}\int_{\mathbb{R}_t\times\mathbb{C}_z\times\mathbb{C}_w}dzdw\mathrm{Tr}\left(AdA+\frac{2}{3}A^3+2\epsilon_2A\{A,A\}\right)+\int_{\{0\}\times\mathbb{C}_z\times\mathbb{C}_w}\beta\bar{\partial}_A\gamma$$
$$-\frac{\epsilon_2'-\epsilon_2}{\epsilon_1}\int_{\mathbb{R}_t^+\times\mathbb{C}_z\times\mathbb{C}_w}\mathrm{Tr}\left(A\wedge\partial A\wedge\partial A\right). \tag{64}$$

Therefore, we can cancel the anomaly of the domain wall by imposing the non-commutativity parameter $\epsilon_2$ on the one side of the wall jumps to a different non-commutativity parameter $\epsilon_2'$ on the other side of the wall:

$$\epsilon_2'=\epsilon_2+n\epsilon_1. \tag{65}$$

Note that $n\sim N$ up to a proportionality constant since the source of the anomaly, which we have just canceled, is the fermions in the $N$ 4d $\mathcal{N}=1$ chiral multiplets.

# 4 Algebra of operators on the domain wall

In this section, we will compute the commutation relation of two generic elements of the algebra of physical operators on the domain wall. In §4.1, we prove that the algebra is a deformation of the universal enveloping algebra of $\mathrm{Diff}_{\epsilon_2}\mathbb{C}\otimes\mathfrak{gl}_K$. For a concise presentation, we simply summarize the idea here. The reader who is interested in the detail of the calculation may refer to Appendix B. The computation is only valid in a careful large $N$ limit and this is implicitly assumed in §4.1. We explain how we took the large N limit in some detail in §4.2. Finally, we conjecture the universal enveloping algebra that we discovered is the classical part of its unique deformation $U_{\epsilon_1}(\mathrm{Diff}_{\epsilon_2}\mathbb{C}\otimes\mathfrak{gl}_K)$, which coincides with the tree-level part of the bulk algebra in §4.3,

## 4.1 The commutation relation

Now, we can compute the commutator of the physical observables. We will prove that the mode algebra is the universal enveloping algebra of $U(\mathrm{Diff}_{\epsilon_2}\mathbb{C}\otimes\mathfrak{gl}_K)$. Consider

$$\left[\mathcal{C}(z^mw^nT^A),\mathcal{C}(z^rw^sT^B)\right]=$$
$$\left[\int_{S_0^3(z_1,w_1)}dz_1dw_1J_1^{(1)}\left(z_1^mw_1^nT^A\right),\int_{S_0^3(z_2,w_2)}dz_2dw_2J_2^{(1)}\left(z_2^rw_2^sT^B\right)\right]. \tag{66}$$

We can compute the commutator[12] by contracting pairs of free fields that give nonzero values. The nonzero propagators that appear in the computation are

$$\langle\partial\phi\bar{\phi}\rangle,\quad\langle\psi_+\bar{\psi}_+\rangle,\quad\langle\psi_+\bar{\psi}_-\rangle. \tag{67}$$

There are two sorts of terms in the RHS: terms from a single contraction and terms from double contractions. As shown in detail in the first part of Appendix B, the single contracted terms reorganize into

$$\mathcal{C}\left([z^mw^nT_A,z^rw^sT_B]\right), \tag{68}$$

where we used the $S^3$ residue integral in the computation

$$\int_{S_{z_0,w_0}^3(z_1,w_1)}\omega_{BM}(z_1-z_0,w_1-w_0)f(z_1,w_1)=f(z_0,w_0). \tag{69}$$

---

[12]Note that the product used in the above commutator is not a secondary product, since the one we are discussing is the product between two descent operators.

A few remarks follow. One may wonder why there is no gauge coupling $g$ dependence[13] in (68), since there can be gauge boson loop corrections to the propagators (67) that we use in the computation. The reason that we do not see such corrections is that the Yang-Mills terms that provide the gauge coupling are Q-exact[14] in our set-up. Therefore, all $g-$dependence is Q-exact.

To compute the double contraction, it is useful to rewrite the integral contour, appealing to the usual 2d CFT $S^1$ contour rearrangement used in the computation of mode algebra in the affine Kac-Moody VOA. We perform the following change of $S^3$ contours

$$\left[S^3_{z_0}(z_1)\right]\left[S^3_{z_0}(z_2)\right] - \left[S^3_{z_0}(z_2)\right]\left[S^3_{z_0}(z_3)\right] = \left[S^3_{z_0}(z_2)\right]\left[S^3_{z_2}(z'_1)\right]. \tag{70}$$

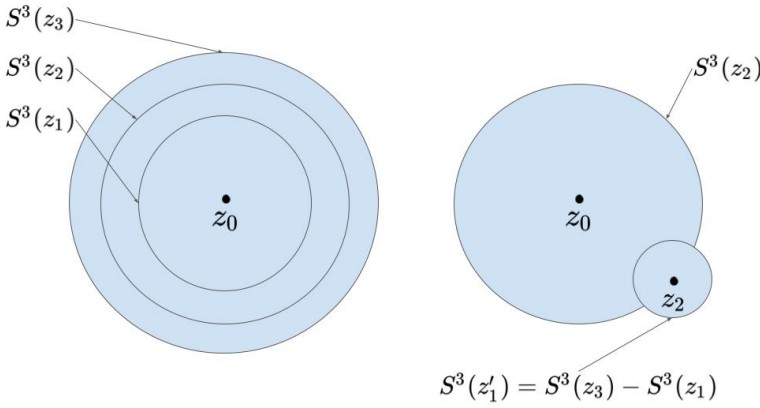

Figure 3: The $S^3$ contour rearrangement. $S^3(z_i)$ is a 3-sphere embedded in $\mathbb{C}_z \times \mathbb{C}_w$ and parametrized by $z_i$ and $w_i$.

Using this contour and performing the double contractions in (66), we end up with

$$\begin{aligned}
&\int_{S^3_0(z_2,w_2)} \int_{S^3_{z_2,w_2}(z_1,w_1)} \omega_{BM}(z_1-z_2,w_1-w_2)(\bar{w}_{12}d\bar{z}_2 - \bar{z}_{12}d\bar{w}_2)z_1^m w_1^n z_2^r w_2^s \operatorname{tr} T^A T^B \\
&= \int_{S^3_0(z_2)} (0)z_2^{m+r} w_2^{n+s} \operatorname{tr} T^A T^B \\
&= 0.
\end{aligned} \tag{71}$$

We wrote down the details of the double contractions in the second part of Appendix B. Therefore, we get

$$\left[\mathcal{C}(z^m w^n T^A), \mathcal{C}(z^r w^s T^B)\right] = \mathcal{C}([z^m w^n T^A, z^r w^s T^B]), \tag{72}$$

where the commutator in the argument of $\mathcal{C}(\bullet)$ in the RHS is that of $\mathfrak{gl}_K \otimes \operatorname{Diff}_{\epsilon_2}\mathbb{C}$. From this, we see that the domain wall algebra of operators satisfies the commutation relations of the algebra $U(\mathfrak{gl}_K \otimes \operatorname{Diff}_{\epsilon_2}\mathbb{C})$.

Before closing this subsection, we would like to point out that Johansen derived the 2d $W-$algebra, using second-type operators in 4d $\mathcal{N} = 1$ holomorphic field theory on $\mathcal{M}_4^H = \Sigma_z \times \mathbb{C}_w^\times$ in [9]. Here $\Sigma_z$ is a Riemann surface parametrized by $z$ and $\mathbb{C}_w^\times$ is a punctured complex plane parametrized by $w$. The second-type operator that Johansen used is "2-dimensional", different from our "3-dimensional" second-type operator, as he integrated

---

[13]We are thankful to Christopher Beem and Kevin Costello for discussion on this point.

[14]Here, the relevant $Q$ is the supercharge that defines the topological holomorphic sector of the 7d SYM on the worldvolume of the $N$ D6' branes.

the first descent operator over the Riemann surface $\Sigma_z$ by wedging it with a holomorphic top form $dz$ of $\Sigma_z$:

$$\int_{\Sigma_z \times \{w\}} dz \wedge \mathbf{Q}\mathcal{O}. \tag{73}$$

Since this operator is a point on $\mathbb{C}_w^\times$ and "wraps" $\Sigma_z$, when two such operators approach in $\mathbb{C}_w^\times$, we can expect the meromorphic structure in the OPE. By effectively[15] reducing the spacetime dimension to 2d via the second-type operator, Johansen avoided Hartog's theorem in 4d.

We can see a hint of the 2d $\mathcal{W}_{K+\infty}$ algebra from a bare form of our universal enveloping algebra. Taking $\mathcal{M}_4^H = \Sigma \times \mathbb{C}^\times = \mathbb{C} \times \mathbb{C}^\times$, and inserting it into our universal enveloping algebra $U(\mathfrak{gl}_K \otimes \mathcal{O}_{\epsilon_2}(\mathcal{M}_4^T))$, we obtain $U(\mathfrak{gl}_K \otimes \mathbb{C}_{\epsilon_2}[z, w, w^{-1}])$. Setting $\epsilon_2 = 1$, we almost get the universal enveloping algebra, which is one of the definition of the $\mathcal{W}_{K+\infty}$ algebra [16]. However, importantly, we do not reproduce the central extension of $\mathcal{W}_{K+\infty}$ in this way, since our commutation relation lacks the central extension due to the vanishing double contraction. It would be interesting to clarify the relation between Johansen's algebra and our algebra.

## 4.2 The large N limit

Trying to avoid potential confusion, we have not discussed the large $N$ limit yet. Let us describe how to take the large $N$ limit and explain the implicit assumption that we make in the previous subsection. We will mostly follow [16], where the author explained how to take the large $N$ limit of the algebra of operators living on $N$ M5 branes. The difference is that we do not need supergroups to proceed.

Let us denote by $Obs_{\epsilon_2}^{4d}(N)$ the algebra of gauge invariant operators on the domain wall coming from a stack of $N$ D6' branes. For $N' \geq N$, there is a natural block diagonal embedding $\mathfrak{gl}_N \hookrightarrow \mathfrak{gl}_{N'}$, which gives rise to a map between algebras

$$\rho_N^{N'} : Obs_{\epsilon_2}^{4d}(N') \to Obs_{\epsilon_2}^{4d}(N). \tag{74}$$

It maps the generator $\mathcal{C}(z^r w^s T^A)$ to its restriction to $\mathfrak{gl}_N$ fields. Note that $\rho_N^{N'}$ is a homomorphism of algebras since the computation of commutation relations in the last section holds for all $N$. $\rho_N^{N'}$ is also surjective as generators are mapped to generators. Therefore, we see that the map defined in the last section $U(\text{Diff}_{\epsilon_2}\mathbb{C} \otimes \mathfrak{gl}_K) \to Obs_{\epsilon_2}^{4d}(N)$ is compatible with the transition map $\rho_N^{N'}$. The only relations are trace relations and by sending $N \to \infty$ all trace relations are gone. Hence, the map $U(\text{Diff}_{\epsilon_2}\mathbb{C} \otimes \mathfrak{gl}_K) \to Obs_{\epsilon_2}^{4d}(N)$ becomes an isomorphism in the large $N$ limit.

## 4.3 Conjecture

Let us discuss the implication of the result that we obtained. We proved that the algebra of operators living on the domain wall is the universal enveloping algebra of $\text{Diff}_{\epsilon_2}\mathbb{C} \times \mathfrak{gl}_K$. It is important to notice that this algebra has no $\epsilon_1$ parameter. Classically, since both the 4d VOA and the boundary condition of the 5d gauge field that were used to construct the set of generators of the enveloping algebra do not depend on $\epsilon_1$, the resulting algebra constructed out of these ingredients should not depend on $\epsilon_1$. Still, there can be potential quantum corrections that depend on $\epsilon_1$ at the level of both generators and relations, similar to the M5 brane example [16]. The quantum correction is reflected in the change of the original structure constants of the algebra $U(\text{Diff}_{\epsilon_2}\mathbb{C} \otimes \mathfrak{gl}_K)[[\epsilon_1]]$ to a new one, i.e. this is a flat deformation

---

[15]See also [48–50] for another example, where the meromorphic structure naturally arises in 4d $\mathcal{N} = 2$ superconformal field theory without relying on the second-type operators.

of $U(\mathrm{Diff}_{\epsilon_2}\mathbb{C}\otimes\mathfrak{gl}_K)$ over the base ring $\mathbb{C}[[\epsilon_1]]$. Let us denote this deformed algebra as $\mathcal{A}_{\epsilon_1}^{KK}$. Then, what is this algebra?

It is in general a hard question to characterize a deformation of the universal enveloping algebra of a certain Lie algebra, but this particular case of interest was explicitly worked out in the $N$ M2 brane twisted holography [17]. One important result of [17] is the uniqueness of the deformation of $U(\mathrm{Diff}_{\epsilon_2}\mathbb{C}\otimes\mathfrak{gl}_K)$.[16] Now, recall from §2.1 that $Obs_{\epsilon_1,\epsilon_2}^{5d\ CS}$ the algebra of operators in the 5d Chern-Simons theory is Koszul dual of $U_{\epsilon_1}(\mathrm{Diff}_{\epsilon_2}\mathbb{C}\otimes\mathfrak{gl}_K)$. Since $U_{\epsilon_1}(\mathrm{Diff}_{\epsilon_2}\mathbb{C}\otimes\mathfrak{gl}_K)$ is the unique deformation of $U(\mathrm{Diff}_{\epsilon_2}\mathbb{C}\otimes\mathfrak{gl}_K)$, $\mathcal{A}_{\epsilon_1}^{KK}$ must be isomorphic to $U_{\epsilon_1}(\mathrm{Diff}_{\epsilon_2}\mathbb{C}\otimes\mathfrak{gl}_K)$. This leads us to conjecture[17] that the quantum deformation of our classical result is Koszul dual to $Obs_{\epsilon_1,\epsilon_2}^{5d\ CS}$ up to a coordinate change $\epsilon_1 \to \epsilon_1 + f_2(\kappa)\epsilon_1^2 + f_3(\kappa)\epsilon_1^3 + \cdots$, where $f_i(\kappa)\in\mathbb{C}[\kappa]$ are polynomials of $\kappa$, establishing the conjectural twisted holography for the domain wall example.

Let us give more details about the unique deformation of the enveloping algebra. Even though the deformation of the algebra is uniquely fixed, we cannot immediately deduce the algebraic relations of the deformed algebra. This algebra actually appeared in another example of twisted M-theory. In [17], the author considered $N$ M2 branes in the twisted M-theory background. Similar to our case, the twisted M-theory background localizes to the 5d Chern-Simons theory, whose algebra of operators is again $Obs_{\epsilon_1,\epsilon_2}^{5d\ CS}$. The worldvolume theory of the $N$ M2 branes is the 3d $\mathcal{N}=4$ $G=U(N)$ gauge theory with $K$ fundamental hypermultiplets and an adjoint hypermultiplet with Rozansky-Witten twist and $\Omega$-deformation applied to two directions out of the 3d worldvolume. The author computed $\mathcal{A}_{\epsilon_1,\epsilon_2}$ the algebra of operators on the $N$ M2 branes and drew an isomorphism between $\mathcal{A}_{\epsilon_1,\epsilon_2}$ and the Koszul dual algebra of $Obs_{\epsilon_1,\epsilon_2}^{5d\ CS}$. Hence, we can equally understand $U_{\epsilon_1}(\mathrm{Diff}_{\epsilon_2}\otimes\mathfrak{gl}_K)$ as $\mathcal{A}_{\epsilon_1,\epsilon_2}$. The generators and relations of this algebra can be found in [23, 24].

The conjectural isomorphism between the quantum deformation of the algebra of operators on the domain wall $U_{\epsilon_1}(\mathrm{Diff}_{\epsilon_2}\mathbb{C}\otimes\mathfrak{gl}_K)$ and the Koszul dual of $Obs_{\epsilon_1,\epsilon_2}^{5d\ CS}$ leads to the statement of the twisted holography of the domain wall in twisted M-theory. It is rather surprising that the domain wall algebra is related to the M2 brane algebra $A_{\epsilon_1,\epsilon_2}$, since there is no obvious structural similarity between the two quantum field theories on the M2 brane and the KK monopole intersection. It would be interesting to figure out a more clear physical reason.

# 5 Conclusion and open questions

In this paper, we study the last available BPS defect in the twisted M-theory, a domain wall-like defect, building on the already established line and surface-like defects in the literature. We identified the algebra of physical operators on the domain wall and establish the isomorphism between the boundary algebra and the bulk operator algebra, which is the algebra of operators of the 5d Chern-Simons theory. The higher vertex operator algebra, which is the algebraic structure under the holomorphically twisted 4d $\mathcal{N}=1$ theory, plays a key role in the derivation.

We engineered a domain wall-like defect in the 5d Chern-Simons theory by introducing a new KK monopole in the twisted M-theory. The new KK monopole along with the already existing Taub-NUT geometry forms a new $G_2$-holonomy 7-manifold, which is topologically twisted. In the type IIA frame, the singular $G_2$-holonomy manifold is mapped to an intersecting D6-D6' brane configuration. The D6-D6' strings are localized at the 4d intersection, where the

---

[16]More precisely, the deformation space of $U(\mathrm{Diff}_{\epsilon_2}\mathbb{C}\otimes\mathfrak{gl}_{K+R|R})$ for all $R$ in a compatible way is isomorphic to $\mathbb{C}[\kappa]$, where $\kappa$ is the central element $1\otimes\mathrm{Id}_{K+R|R}$.

[17]As we have said in the introduction, we believe this conjecture is true a priori. However, since we do not know how to compute the quantum corrections using the defining properties of the higher VOA, not relying on Costello's theorem, we modestly call it as a conjecture.

holomorphic twist is applied. The intrinsic degree of freedom living on the wall is the 4d $\mathcal{N} = 1$ bi-fundamental chiral multiplet. After passing to the Q-cohomology, specified by the holomorphic twist, one gets a higher version of the vertex operator algebra, which has a concise presentation as the 4d $\beta\gamma$ system. The algebra of operators in the 4d $\beta\gamma$ system is generated by the integrated higher current. Combining it with the boundary condition of the 5d gauge field on the domain wall, we form a set of generators of the physical operators living on the domain wall. We prove that this algebra is the universal enveloping algebra of $\mathrm{Diff}_{\epsilon_2}\mathbb{C} \otimes \mathfrak{gl}_K$ and conjecture an isomorphism with the Koszul dual of the bulk algebra using the uniqueness of deformation of $U(\mathrm{Diff}_{\epsilon_2}\mathbb{C} \otimes \mathfrak{gl}_K)$.

We can summarize the operator algebras on M2, M5, and the new KK monopole in the twisted M-theory on $\mathbb{R}_t \times \mathbb{C}_z \times \mathbb{C}_w \times \mathbb{C}_{\epsilon_1} \times \mathrm{TN}_K$ in the following table.

Table 1: Summary of BPS defects, associated field theory, twists, protected subsectors of the field theory, and algebras. ADHM means the $G = U(N)$ gauge theory with 1 adjoint hypermultiplet and $K$ fundamental hypermultiplets. T and H stand for topological and holomorphic twists. Field theories I and II are the original theories of the relevant branes and their protected subsectors. TQM is an abbreviation of topological quantum mechanics.

| BPS defect | Field theory I | Twist | Field theory II | Algebra |
|---|---|---|---|---|
| M2 | 3d $\mathcal{N} = 4$ ADHM | 3T | 1d TQM | $U_{\epsilon_1}(\mathrm{Diff}_{\epsilon_2}\mathbb{C} \otimes \mathfrak{gl}_K)$ |
| M5 | 6d $\mathcal{N} = (2,0)$ | 4T, 2H | 2d fermions | $U_{c=N}(\mathcal{O}_{\epsilon_2}(\mathbb{C} \times \mathbb{C}^\times) \otimes \mathfrak{gl}_K)$ |
| KKM | 4d $\mathcal{N} = 1$ chiral | 3T, 4H | 4d $\beta\gamma$ system | $U_{\epsilon_1}(\mathrm{Diff}_{\epsilon_2}\mathbb{C} \otimes \mathfrak{gl}_K)$ |

There is a notable difference between the last example and the first two examples. Although the first two examples have an algebra of operators constructed solely from the first-type operators, in other words, local operators in the Q-cohomology, the algebra of operators of the last example involves the second-type operators, in other words, $\mathbf{Q}-$descendant operators.

On the other hand, there is a similarity shared by all three examples: the effective codimension of the elements of algebras is one. Let us explain more about what we mean by the effective codimension. First, in 1d TQM we easily see that the codimension of a local operator is 1. Second, in 2d VOA, when one computes the mode algebra, one extracts a mode from a current operator by using $S^1$ residue integral. The codimension of the associated $S^1$ in 2d spacetime is 1. Lastly, for higher VOA, when we derive the mode algebra on the domain wall, we used $S^3$ contour to extract elements of the algebra. The non-commutative structure comes from the codimension-1 property of $S^3$ in 4d spacetime.

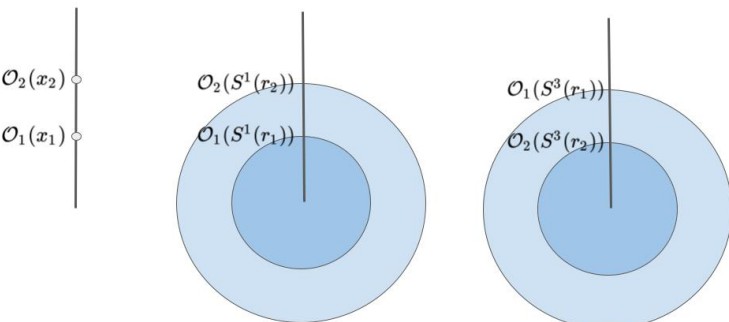

Figure 4: From left to right, 1d TQM, 2d VOA, and 4d VOA operators, depicted by the circles. The thick straight lines are the transverse directions of the operators, showing the codimension-1 nature of the algebra of operators in each case.

Let us now list some open questions that we hope to answer in the future:

- What would be the canonical Koszul-dual pair interaction between the 5d gauge theory and 4d holomorphic field theory at the level of modes? Can we derive a deformed version of the commutation relation (72) by imposing the BRST invariance of a certain set of Feynman diagrams, like [17]? It is not clear to us how to define a local interaction between a ghost and a second-type operator. Once it is identified, the computation will be as straightforward as that of [17, 24].

- Can we identify a supersymmetric or at least stable configuration of the combined system of domain wall, line, and surface defects? [30, 31] studied the network of M2 and M5 branes and identified the relevant fusion algebra using free field realization of the M2 and M5 brane algebras and Feynman diagram computation in 5d Chern-Simons theory. Incorporating the domain wall defect in this line of analysis seems to be a very natural direction to pursue.

- In [52], Li and Yamazaki gave a complete answer for algebras acting on BPS partition functions that count D6-D2-D0 bound states on toric Calabi-Yau 3-folds. We wonder if there is a connection between the algebra associated to our 5d/4d/5d system and a potential BPS partition function that counts M2 branes wrapping associative 3-cycles in the $G_2$ manifold.

## Acknowledgements

We thank Christopher Beem, Kevin Costello, and Junya Yagi for illuminating discussions. We especially thank Kevin Costello for his insightful comments on our draft. Finally, we are grateful to two anonymous SciPost referees who gave us valuable remarks on various parts of the first version of this paper.

**Funding information**   Research of JO was supported by ERC Grants 682608 and 864828. Research at the Perimeter Institute is supported by the Government of Canada through Industry Canada and by the Province of Ontario through the Ministry of Economic Development & Innovation.

## A   4d $\mathcal{N} = 1$ SUSY transformation of a chiral multiplet

In this appendix, we summarize the supersymmetry transformation of the 4d $\mathcal{N} = 1$ chiral multiplet $\Phi = (\phi, \psi_\alpha, F)$ and its conjugate $\bar{\Phi} = (\bar{\phi}, \bar{\psi}_{\dot{\alpha}}, \bar{F})$. In particular, we only record the transformations, generated by the scalar $Q$, which we used to define the Q-cohomology, and the 1-form supercharge $\mathbf{Q}$, which we used to define the second-type operators in the Q-

cohomology:

$$
\begin{aligned}
Q\phi &= \psi_-, \\
Q\psi_+ &= -F, \\
Q\psi_- &= 0, \\
QF &= 0, \\
Q\bar{\phi} &= 0, \\
Q\bar{\psi}_+ &= -i\partial_{\bar{z}}\bar{\phi}, \\
Q\bar{\psi}_- &= i\partial_{\bar{w}}\bar{\phi}, \\
Q\bar{F} &= i\partial_{\bar{w}}\bar{\psi}_+ + i\partial_{\bar{z}}\bar{\psi}_-.
\end{aligned}
\tag{75}
$$

Up to on-shell condition, there are three Q-closed operators:

$$
\psi_+, \quad \psi_-, \quad \bar{\phi}.
\tag{76}
$$

Since $\psi_-$ is Q-exact by the first equation, the operators in the Q-cohomology are

$$
\bar{\phi}, \quad \psi_+
\tag{77}
$$

and their holomorphic derivatives with respect to $\partial_z$, $\partial_w$.

Next, we record the action of 1-form supercharge on the on-shell chiral multiplet fields:

$$
\begin{aligned}
\mathbf{Q}\phi &= 0, \\
\mathbf{Q}\psi_+ &= d\bar{z}\,\partial_w\phi - d\bar{w}\,\partial_z\phi, \\
\mathbf{Q}\psi_- &= d\bar{z}\,\partial_{\bar{z}}\phi + d\bar{w}\,\partial_{\bar{w}}\phi, \\
\mathbf{Q}\bar{\phi} &= id\bar{z}\,\bar{\psi}_+ - id\bar{w}\,\bar{\psi}_-, \\
\mathbf{Q}\bar{\psi}_+ &= -id\bar{w}\bar{F}, \\
\mathbf{Q}\bar{\psi}_- &= -id\bar{z}\bar{F}.
\end{aligned}
\tag{78}
$$

Since we only use the action of 1-form supercharge on (77), only second and fourth lines are meaningful in the main text of this paper.

## B Mode algebra computation

In this appendix, we will provide the details on the commutator computation summarized in §4.1.

$$
\left[\mathcal{C}(z^m w^n T^A), \mathcal{C}(z^r w^s T^B)\right] = \mathcal{C}\left(\left[z^m w^n T^A, z^r w^s T^B\right]\right) + 0,
\tag{79}
$$

where

$$
\begin{aligned}
\mathcal{C}(z^m w^n T^A) &= \int_{S_0^3(z_1, w_1)} dz_1 dw_1 J_1^{(1)}(z_1^m w_1^n T^A), \\
\mathcal{C}(z^r w^s T^B) &= \int_{S_0^3(z_2, w_2)} dz_2 dw_2 J_2^{(1)}(z_2^r w_2^s T^B).
\end{aligned}
\tag{80}
$$

We will first focus on expanding $J_1^{(1)}$ and $J_2^{(1)}$ and identify the nonzero propagators that we will contract later:

$$
\begin{aligned}
J_1^{(1)} J_2^{(1)} = & \left[(\bar{\psi}_- d\bar{z}_1 - \bar{\psi}_+ d\bar{w}_1)\psi_+ + \bar{\phi}(\partial_{w_1}\phi\, d\bar{z}_1 - \partial_{z_1}\phi\, d\bar{w}_1)\right]_1 \\
& \cdot \left[(\bar{\psi}_- d\bar{z}_2 - \bar{\psi}_+ d\bar{w}_2)\psi_+ + \bar{\phi}(\partial_{w_2}\phi\, d\bar{z}_2 - \partial_{z_2}\phi\, d\bar{w}_2)\right]_2.
\end{aligned}
\tag{81}
$$

Out of 16 terms in the expansion, 8 terms can host nonzero contractions. Those are

$$
\begin{aligned}
& d\bar{z}_1 d\bar{z}_2 \big[(\bar\psi_-\psi_+)_1(\bar\psi_-\psi_+)_2 + (\bar\phi\,\partial_{w_1}\phi)_1(\bar\phi\,\partial_{w_2}\phi)_2\big] \\
& + d\bar{w}_1 d\bar{w}_2 \big[(-\bar\psi_+\psi_+)_1(-\bar\psi_+\psi_+)_2 + (-\bar\phi\,\partial_{z_1}\phi)_1(-\bar\phi\,\partial_{z_2}\phi)_2\big] \\
& + d\bar{z}_1 d\bar{w}_2 \big[(\bar\psi_-\psi_+)_1(-\bar\psi_+\psi_+)_2 + (\bar\phi\,\partial_{w_1}\phi)_1(-\bar\phi\,\partial_{z_2}\phi)_2\big] \\
& + d\bar{w}_1 d\bar{z}_2 \big[(-\bar\psi_+\psi_+)_1(\bar\psi_-\psi_+)_2 + (-\bar\phi\,\partial_{z_1}\phi)_1(\bar\phi\,\partial_{w_2}\phi)_2\big].
\end{aligned}
\tag{82}
$$

Now, we will use

$$
\begin{aligned}
\langle \partial_{z_i}\phi_i(z_i,w_i)\bar\phi_j(z_j,w_j)\rangle &= \frac{\bar{z}_{ij}}{d_{ij}^2}, \quad
\langle \partial_{w_i}\phi_i(z_i,w_i)\bar\phi_j(z_j,w_j)\rangle = \frac{\bar{w}_{ij}}{d_{ij}^2}, \\
\langle \bar\psi_{+,i}(z_i,w_i)\psi_{+,j}(z_j,w_j)\rangle &= \frac{\bar{z}_{ij}}{d_{ij}^2}, \quad
\langle \bar\psi_{-,i}(z_i,w_i)\psi_{+,j}(z_j,w_j)\rangle = \frac{\bar{w}_{ij}}{d_{ij}^2},
\end{aligned}
\tag{83}
$$

where

$$
d_{ij}^2 = |z_{ij}|^2 + |w_{ij}|^2,
\tag{84}
$$

in the following.

## Single contraction

For each term in (82) that looks

$$
(AB)_1(AB)_2,
\tag{85}
$$

we may contract either $\langle A_1 B_2\rangle$ or $\langle B_1 A_2\rangle$. Summing up all possible single contractions, we get

$$
\begin{aligned}
\frac{1}{d_{12}^2}\Big( & d\bar{z}_1 d\bar{z}_2 \bar{w}_{12}(\psi_{+,1}\bar\psi_{-,2} + \bar\psi_{-,1}\psi_{+,2} + \partial_{w_1}\phi_1\bar\phi_2 + \bar\phi_1\partial_{w_2}\phi_2) \\
& + d\bar{w}_1 d\bar{w}_2 \bar{z}_{12}(\psi_{+,1}\bar\psi_{+,2} + \bar\psi_{+,1}\psi_{+,2} + \partial_{z_1}\phi_1\bar\phi_2 + \bar\phi_1\partial_{z_2}\phi_2) \\
& - d\bar{z}_1 d\bar{w}_2 (\bar{w}_{12}\psi_{+,1}\bar\psi_{+,2} + \bar{z}_{12}\bar\psi_{-,1}\psi_{+,2} + \bar{z}_{12}\partial_{w_1}\phi_1\bar\phi_2 + \bar{w}_{12}\bar\phi_1\partial_{z_2}\phi_2) \\
& - d\bar{w}_1 d\bar{z}_2 (\bar{z}_{12}\psi_{+,1}\bar\psi_{-,2} + \bar{w}_{12}\bar\psi_{+,1}\psi_{+,2} + \bar{w}_{12}\partial_{z_1}\phi_1\bar\phi_2 + \bar{z}_{12}\bar\phi_1\partial_{w_2}\phi_2) \Big).
\end{aligned}
\tag{86}
$$

Next, rearrange the above so that we can find the following $S^3$ integral kernel

$$
\omega_{BM}(z,w) = \frac{\bar{w}d\bar{z} - \bar{z}d\bar{w}}{(|z|^2 + |w|^2)^2}
\tag{87}
$$

manifestly

$$
\begin{aligned}
& d\bar{z}_1 \frac{\bar{w}_{12}d\bar{z}_2 - \bar{z}_{12}d\bar{w}_2}{d_{12}^2}\bar\psi_{-,1}\psi_{+,2} + d\bar{w}_1 \frac{\bar{z}_{12}d\bar{w}_2 - \bar{w}_{12}d\bar{z}_2}{d_{12}^2}\bar\psi_{+,1}\psi_{+,2} \\
& + d\bar{z}_1 \frac{\bar{w}_{12}d\bar{z}_2 - \bar{z}_{12}d\bar{w}_2}{d_{12}^2}\partial_{w_1}\phi_1\bar\phi_2 + d\bar{w}_1 \frac{\bar{z}_{12}d\bar{w}_2 - \bar{w}_{12}d\bar{z}_2}{d_{12}^2}\partial_{z_1}\phi_1\bar\phi_2 \\
& + d\bar{z}_2 \frac{\bar{w}_{12}d\bar{z}_1 - \bar{z}_{12}d\bar{w}_1}{d_{12}^2}\psi_{+,1}\bar\psi_{-,2} + d\bar{w}_2 \frac{\bar{z}_{12}d\bar{w}_1 - \bar{w}_{12}d\bar{z}_1}{d_{12}^2}\psi_{+,1}\bar\psi_{+,2} \\
& + d\bar{z}_2 \frac{\bar{w}_{12}d\bar{z}_1 - \bar{z}_{12}d\bar{w}_1}{d_{12}^2}\bar\phi_1\partial_{w_2}\phi_2 + d\bar{w}_2 \frac{\bar{z}_{12}d\bar{w}_1 - \bar{w}_{12}d\bar{z}_1}{d_{12}^2}\bar\phi_1\partial_{z_2}\phi_2.
\end{aligned}
\tag{88}
$$

Then, make the following change of variables

$$
(z_2, w_2) \to (\tilde{z}_2, \tilde{w}_2) = (z_2 - z_1, w_2 - w_1)
\tag{89}
$$

in the first two lines of (88) to get $\omega_{BM}(\tilde{z}_2, \tilde{w}_2)$ and make the following change of variables

$$(z_1, w_1) \rightarrow (\tilde{z}_1, \tilde{w}_1) = (z_1 - z_2, w_1 - w_2) \tag{90}$$

in the last two lines of (88) to get $\omega_{BM}(\tilde{z}_1, \tilde{w}_1)$. We have expanded $J_1^{(1)} \cdot J_2^{(1)}$. Let us plug this back in

$$\mathcal{C}(z^m w^n T^A) \cdot \mathcal{C}(z^r w^s T^B). \tag{91}$$

We will omit the elements of $\mathbb{C}[z, w] \otimes \mathfrak{gl}_K$ and restore those at the end.

Then, we can rearrange (91) into

$$\int_{S_0^3} \omega_{BM}(\tilde{z}_2, \tilde{w}_2) \int_{S_0^3} dz_1 dw_1 \left( (\bar{\psi}_{-,1} d\bar{z}_1 - \bar{\psi}_{+,1} d\bar{w}_1) \psi_{+,2} + \bar{\phi}_2 (\partial_{w_1} \phi_1 d\bar{z}_1 - \partial_{z_1} \phi_1 d\bar{w}_1) \right)$$

$$+ \int_{S_0^3} \omega_{BM}(\tilde{z}_1, \tilde{w}_1) \int_{S_0^3} dz_2 dw_2 \left( (\bar{\psi}_{-,2} d\bar{z}_2 - \bar{\psi}_{+,2} d\bar{w}_2) \psi_{+,1} + \bar{\phi}_1 (\partial_{w_2} \phi_2 d\bar{z}_2 - \partial_{z_2} \phi_2 d\bar{w}_2) \right), \tag{92}$$

where $S_0^3$ is a 3-sphere centered at the origin of $\mathbb{C}_z \times \mathbb{C}_w$. Note that $\psi_{+,2}$ and $\bar{\phi}_2$ in the first line of (92) are functions of $\tilde{z}_2 + z_1$ and $\tilde{w}_2 + w_1$ due to (89). Similarly, $\psi_{+,1}$ and $\bar{\phi}_1$ are functions of $\tilde{z}_1 + z_2$ and $\tilde{w}_1 + w_2$ due to (90). Therefore, so far, the integrands are bi-local functions on two sets of variables. Evaluating the first $S^3$ residue integrals in each line of (92), we get

$$\int_{S_0^3} dz_1 dw_1 \left( (\bar{\psi}_{-,1} d\bar{z}_1 - \bar{\psi}_{+,1} d\bar{w}_1) \psi_{+,1} + \bar{\phi}_1 (\partial_{w_1} \phi_1 d\bar{z}_1 - \partial_{z_1} \phi_1 d\bar{w}_1) \right)$$

$$+ \int_{S_0^3} dz_2 dw_2 \left( (\bar{\psi}_{-,2} d\bar{z}_2 - \bar{\psi}_{+,2} d\bar{w}_2) \psi_{+,2} + \bar{\phi}_2 (\partial_{w_2} \phi_2 d\bar{z}_2 - \partial_{z_2} \phi_2 d\bar{w}_2) \right). \tag{93}$$

Due to the residue integrals, all fields with subscript $i$ now depend on a single set of variables $(z_i, w_i)$, so there is no bilocal behavior anymore. Therefore, we observe that each integrand corresponds to $J_i^{(1)}$ and rewrite (93) as

$$\int_{S^3} dz_1 dw_1 J_1^{(1)} + \int_{S^3} dz_2 dw_2 J_2^{(1)}. \tag{94}$$

Since the subscripts are now dummy variables, we can unify (94) as one term. After restoring the omitted elements of $\mathbb{C}[z, w] \otimes \mathfrak{gl}_K$, we get

$$\int_{S^3} dz dw J^{(1)} \left( z^m w^n T^A \right) \left( z^r w^s T^B \right). \tag{95}$$

This is the result of the single contractions of a product of two algebra generators (91). To get the commutator, we permute two elements of $\mathbb{C}[z, w] \otimes \mathfrak{gl}_K$ in (95) and subtract the resulting term from (95). We get

$$\int_{S^3} dz dw J^{(1)} \left[ z^m w^n T^A, z^r w^s T^B \right] =: \mathcal{C}\left( \left[ z^m w^n T^A, z^r w^s T^B \right] \right). \tag{96}$$

It reproduces the first term in the RHS of (79).

## Double contractions

Let us start from the single-contracted expression (88) and perform another contraction:

$$
\frac{1}{d_{12}^4}\Big( d\bar{z}_1(\bar{w}_{12}d\bar{z}_2 - \bar{z}_{12}d\bar{w}_2)\bar{w}_{12} + d\bar{w}_1(\bar{z}_{12}d\bar{w}_{12} - \bar{w}_{12}d\bar{z}_{12})\bar{z}_{12}
$$
$$
+ d\bar{z}_1(\bar{w}_2 d\bar{z}_{12} - \bar{z}_{12}d\bar{w}_2)\bar{w}_{12} + d\bar{w}_1(\bar{z}_{12}d\bar{w}_2 - \bar{w}_{12}d\bar{z}_2)\bar{z}_{12}
$$
$$
+ d\bar{z}_2(\bar{w}_{12}d\bar{z}_1 - \bar{z}_{12}d\bar{w}_1)\bar{w}_{12} + d\bar{w}_2(\bar{z}_{12}d\bar{w}_1 - \bar{w}_{12}d\bar{z}_1)\bar{z}_{12}
$$
$$
+ d\bar{z}_2(\bar{w}_{12}d\bar{z}_1 - \bar{z}_{12}d\bar{w}_1)(-\bar{w}_{12}) + d\bar{w}_{12}(\bar{z}_{12}d\bar{w}_1 - \bar{w}_{12}d\bar{z}_1)(-\bar{z}_{12}) \Big), \tag{97}
$$

where in the third and fourth line we used

$$
\langle \psi_{+,1}\bar{\psi}_{\dot{-},2}\rangle = -\langle\bar{\psi}_{\dot{-},2}\psi_{+,1}\rangle = \langle\bar{\psi}_{\dot{-},1}\psi_{+,2}\rangle, \quad \langle\psi_{+,1}\bar{\psi}_{\dot{+},2}\rangle = -\langle\bar{\psi}_{\dot{+},2}\psi_{+,1}\rangle = \langle\bar{\psi}_{\dot{+},1}\psi_{+,2}\rangle,
$$
$$
\langle\bar{\phi}_1\partial_{w_2}\phi_2\rangle = \langle\partial_{w_2}\phi_2\bar{\phi}_1\rangle = -\langle\partial_{w_1}\phi_1\bar{\phi}_2\rangle, \quad \langle\bar{\phi}_1\partial_{z_2}\phi_2\rangle = \langle\partial_{z_2}\phi_2\bar{\phi}_1\rangle = -\langle\partial_{z_1}\phi_1\bar{\phi}_2\rangle. \tag{98}
$$

We can simplify (97) into

$$
2(\bar{w}_{12}d\bar{z}_2 - \bar{z}_{12}d\bar{w}_2)\frac{\bar{w}_{12}d\bar{z}_1 - \bar{z}_{12}d\bar{w}_1}{d_{12}^4}. \tag{99}
$$

By integrating (99) along the modified contour depicted in Figure 3, we can finish the computation for the double contracted term of (79):

$$
2\int_{S_0^3(z_2,w_2)}\int_{S_{z_2,w_2}^3(z_1,w_1)} \frac{\bar{w}_{12}d\bar{z}_1 - \bar{z}_{12}d\bar{w}_1}{d_{12}^4}(\bar{w}_{12}d\bar{z}_2 - \bar{z}_{12}d\bar{w}_2)(z_1^m w_1^n z_2^r w_2^s \operatorname{tr} T^A T^B)
$$
$$
= 2\int_{S_0^3(z_2,w_2)}\int_{S_{z_2,w_2}^3(z_1,w_1)} \omega_{BM}(z_1-z_2, w_1-w_2)(\bar{w}_{12}d\bar{z}_2 - \bar{z}_{12}d\bar{w}_2)(z_1^m w_1^n z_2^r w_2^s \operatorname{tr} T^A T^B)
$$
$$
= 2\int_{S_0^3(z_1,w_1)} (0)(z_2^m w_2^n z_2^r w_2^s \operatorname{tr} T^A T^B)
$$
$$
= 0, \tag{100}
$$

where $S_{z_i,w_i}^3(z_j,w_j)$ is a 3-sphere centered at $(z_i,w_i)$ and parametrized by $(z_j,w_j)$. In the first equality, we used the fact that $d_{12}^4 = d_{12}^2$ on $S_{z_1,w_1}^3(z_2,w_2)$, assuming the radius of the 3-sphere is 1. The result is independent on the radius, as it should. We conclude that there is no central element in the commutation relation.

Therefore, combining with (96), we recover (79)

$$
\big[\mathcal{C}(z^m w^n T^A), \mathcal{C}(z^r w^s T^B)\big] = \mathcal{C}\big(\big[z^m w^n T^A, z^r w^s T^B\big]\big). \tag{101}
$$

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
