# Peer review of "A domain wall in twisted M-theory"

_SciPost Physics, doi:SciPost Phys. 11, 077 (2021)_

## Round 1 · Referee Report · Anonymous (Referee 1) · 2021-7-29

Strengths
1) The paper provides a conjecture for the twisted holography of a KK monopole in twisted M-theory, which is seemingly the last BPS defect that remained to be considered. Moreover, the conclusions pose further physics questions. 2) The paper uses techniques from recent developments in the formal study of operator algebras of QFT, including higher products.
Weaknesses
1) Despite the presence of a clear introduction, containing a review of the status quaestionis, the paper refers with no explanation to a few technical mathematical results that are probably not familiar to most of the readers. It would be good to at least acknowledge this. 2) The presentation of the paper could be slightly improved.
Report
Requested changes
1) In Section 4.3, the authors find the quantum deformation of the algebra of operators on the domain wall by relying on a previous result by Costello. Whilst in the Introduction it is mentioned that Costello's result is mathematically rigorous, here the authors refer to it both as a proved and a conjectural isomorphism. Could they please clarify this point? More generally, since this is the concluding part of the paper, I would welcome a slower pace in the exposition, stating clearly what is mathematically proved and what is physically conjectured (perhaps in the same vein as the Introduction), especially since twisted holography is an area where results inspired by physics can be rigorously proved algebraically. Finally, it seems curious that the algebras for the M2 and for the KK monopoles are the same, and the authors could consider highlighting this more.
2) The protected subsector for the field theory associated to the KK monopole is described by a field theory in the same dimension as the original one. Is there a physical interpretation for this?
3) Some additional minor clarifications: - p. 9 +9 "massless D6-D6' strings" I think that strings can be tensionless, and they may have massless spectrum. Do the authors mean the latter? - p. 9 -11 as the authors point out, this secondary product is related to the descent procedure introduced by Witten. Here the product is not referred to as "secondary", whereas later on it is without referring to the introduction (p. 11 above (30)). I would suggest introducing the terminology "secondary" at p. 9 or drawing the relation at p. 11. - p. 11 +5 I didn't find this description easy to understand. I would rephrase it as something like "We can construct it by first applying... in the Q-cohomology. The resulting $\mathcal{O}^{(1)}$ is not Q-closed... If we then wedge..." - p. 12 caption to Fig 2. I think that $\mathcal{O}^{(1)}_2$ is a one-form, so it's $\Omega \mathcal{O}^{(1)}_2$ that is supported on $S^3$ - p. 13 below (35) I found the mention of the derived setting a bit off-hand: is that crucial or is that only a nod to the expert reader? - p. 15 -3 Is it correct to say that the boundary conditions in (52) are equivalent? If so, could you mention it explicitly?
4) There are numerous typos and minor grammatical mistakes throughout the paper. Here is a list of some - there should be a space between word and bracket, starting from page 2 line 10 - Kahler and hyperKahler should have a umlaut on the a - p. 2 +14 "Twisted M-theory is an eleven-dimensional..." - p. 2 -12 "M5 branes should wrap a four-cycle in the..." - p. 3 +18 I think it would be clearer as "reducing along a circle in the $G_2$ manifold" (same at p. 5 +3, p. 5 + 15, p. 7 -7, p. 7 -6) - p. 3 -12 something like "We will describe in more detail this set of operators..." (no about), same in p. 11 - 9, p. 23 -2 - p. 4 -14 "squares not to zero but to $\epsilon V$" - p. 4 - 7 "The background induced by $\Psi_\epsilon$..." do the authors mean that it makes the dependence on $\mathcal{M}_7^T$ topological and $\mathcal{M}_4^H$ holomorphic? - p. 4 footnote 2 "Appendix" - p. 5 +6 "7d maximal SYM" - p. 6 -10 "defects cannot engineer" (same elsewhere) - p. 7 -12 "this new geometry $\widetilde{\mathcal{M}}_7^T$" - p. 8 eqn (13) Usually "rotations" have determinant 1: is there a particular reason why the authors took det -1? - p. 10 above (22) "one may act with $\mathbf{Q}$..." - p. 10 -6 "complex one-dimensional holomorphic theory" - p. 10 footnote 7 "distinguish it from the secondary product" - p. 12 below (33) "The Bochner-Martinelli kernel..." - p. 14 -6 "We call $dz \wedge dw \wedge J^{(1)}$ a higher current." - p. 15 -8 " We need to make sure of the locality of the... let us vary the action obtaining... Using Stokes' theorem... boundary conditions which make the boundary variations (51) zero." - p. 16 +2 "valued in the Lie algebra" - p. 18 -6 "The reader who is interested in the detail of the calculation may..." - p. 22 +8 "is reflected in the change of the original structure constants..." - p. 23 +4 "isomorphism between the boundary algebra and" - p. 23 +13 "The intrinsic degree..." - p. 23 -2 "what we mean by..."

Anonymous on 2021-08-13 [id 1671]
Summary:
This paper concerns the derivation of a certain associative algebra in the context of twisted Omega-deformed holography, building on Costello's description of twisted Omega-deformed M theory. A portion of the paper is reviews results of Costello and others, and can possibly be trimmed a bit. The overall emphasis on higher VOA structure is slightly misleading, and needs to be modified (see below). There are some points that need to be addressed further, but overall I think the article is provides a nice contribution to this rapidly evolving field.
Broad comments:
Section 3 is largely expositional, though the authors do make an effort to credit the appropriate sources. A missing source is section 10.5 of https://people.math.umass.edu/~gwilliam/factorization2.pdf. It is in this reference where the free holomorphic OPE of the 4d N=1 theory has already been computed, and they should reference this.
I am also distracted by the emphasis on this higher VOA structure on C^2 being relevant. The main results of the paper pertain to the calculation of a certain *associative algebra* given by the S^3-modes of the 4d system. To argue why the S^3 modes has an algebra structure goes all the way back to the familiar story of `radial quantization' or mode algebra. While radial quantization is related to a conjectural 4d VOA structure (and the authors seem to gesture in this direction), in no way does this 4d VOA structure seem to be important for any computations in this paper. My suggestion would be to remove the discussion of the higher VOA structure unless the authors have some result pertaining to it.
That being said, the paper performs a careful calculation of associative algebra associated to the radial quantization, or mode algebra, of the punctured domain wall. I think this calculation should be the one emphasized in the abstract and introduction, rather than the secondary OPE structure.
Technical comments:
Above equation (23). "The first type of operator is...". What operators are they describing? Also, you mention "Q-cohomology"? What are they considering the Q-cohomology of? Surely not local operators, since they say the second type of operator is an integral over S^3? So they need to clarify what Q-cohomology you are considering.
Equation (30). Does O_i denote a cohomology class or a cochain representative? If the latter, why is this higher OPE well-defined?
Equation (54). This is not a holomorphic function. This is a polynomial function in two algebraic variables. They should emphasize the difference.
Below Equation (78). The sentence "...we get a deformed 4d...". Are they claiming that this derived algebra appears in the M theory setup. They seemed to have just proposed a formula for this algebra, but they did not do any calculation to show how it arises from the theory. Am I missing a step in the calculation? If not, I think they should emphasize that you *expect* this algebra to play a role, rather than make it sound like you have deduced this algebra from the physical theory. It is misleading to introduce this algebra at such a stage in the paper, it should belong in the "conjectural" section or the conclusion.
Typo in Footnote 8. Should be C^D not C^{2D}. Also not sure how useful this footnote is if no further comments or discussion arise later on. Again you can refer to section 10.5 of https://people.math.umass.edu/~gwilliam/factorization2.pdf.

---

## Round 2 · Referee Report · Anonymous (Referee 1) · 2021-9-6

Report

In this resubmitted version, the authors have implemented the referees' suggestions. The paper is therefore recommended for publication without further modifications.

---

## Round 2 · Author Response

Dear referees,

We appreciate your effort to carefully review our paper and gave valuable comments on it.

We have revised the first version following your advice.

Best,
Jihwan

---

## Editorial Decision

published